# Insights into the structure and assembly of a bacterial cellulose secretion system

Petya Violinova Krasteva [1,2,3], Joaquin Bernal-Bayard[4], Laetitia Travier[4,9], Fernando Ariel Martin[4,10], Pierre-Alexandre Kaminski[5], Gouzel Karimova[2,6], Rémi Fronzes[1,2,7,8] & Jean-Marc Ghigo[4]

Secreted exopolysaccharides present important determinants for bacterial biofilm formation, survival, and virulence. Cellulose secretion typically requires the concerted action of a c-di-GMP-responsive inner membrane synthase (BcsA), an accessory membrane-anchored protein (BcsB), and several additional Bcs components. Although the BcsAB catalytic duo has been studied in great detail, its interplay with co-expressed subunits remains enigmatic. Here we show that *E. coli* Bcs proteins partake in a complex protein interaction network. Electron microscopy reveals a stable, megadalton-sized macromolecular assembly, which encompasses most of the inner membrane and cytosolic Bcs components and features a previously unobserved asymmetric architecture. Heterologous reconstitution and mutational analyses point toward a structure–function model, where accessory proteins regulate secretion by affecting both the assembly and stability of the system. Altogether, these results lay the foundation for more comprehensive models of synthase-dependent exopolysaccharide secretion in biofilms and add a sophisticated secretory nanomachine to the diverse bacterial arsenal for virulence and adaptation.

[1] G5 Biologie Structurale de la Sécrétion Bactérienne, Institut Pasteur, 25–28 rue du Docteur Roux, 75015 Paris, France. [2] CNRS, UMR3528, Institut Pasteur, 25–28 rue du Docteur Roux, 75015 Paris, France. [3] Structural Biology of Biofilms Group, Institute for Integrative Biology of the Cell (I2BC), CEA, CNRS, Université Paris Sud, 1 Avenue de la Terrasse, 91198 Gif-sur-Yvette, France. [4] Unité de Génétique des Biofilms, Département de Microbiologie, Institut Pasteur, 25–28 rue du Dr. Roux, 75015 Paris, France. [5] Unité de Biologie des Bactéries Pathogènes à Gram-Positif, Institut Pasteur, 25–28 rue du Docteur Roux, 75015 Paris, France. [6] Unité de Biochimie des Interactions Macromoléculaires, Institut Pasteur, 25–28 rue du Docteur Roux, 75015 Paris, France[7] Structure et Fonction des Nanomachines Bactériennes, Institut Européen de Chimie et Biologie, Rue Robert Escarpit, 33607 Pessac, France. [8] CNRS, UMR5234, Université de Bordeaux, 146 rue Léo Saignat, 33076 Bordeaux, France. [9] Present address: Unité de Biologie des Infections, INSERM U1117, Institut Pasteur, 25–28 rue du Docteur Roux, 75015 Paris, France. [10] Present address: MedILS, Mediterranean Institute for Life Sciences, Meštovićevo šetalište 45, 21000 Split, Croatia. Petya Violinova Krasteva and Joaquin Bernal-Bayard contributed equally to this work. Correspondence and requests for materials should be addressed to P. V.K. (email: petya.krasteva@i2bc.paris-saclay.fr) or to J.-M.G. (email: jmghigo@pasteur.fr)

Bacterial biofilms are collaborative sessile communities embedded in protective extracellular matrix[1]. Biogenesis platforms for the secretion of biofilm matrix components—often controlled directly or indirectly by the intracellular second messenger c-di-GMP[2,3]—are important determinants for biofilm formation, persistence, and bacterial infections[4]. Many Gram-negative bacteria rely on functionally homologous synthase-dependent systems for the secretion of matrix exopolysaccharides[3,5], of which cellulose biogenesis represents a widespread and archetypal c-di-GMP-responsive biofilm effector system[6,7]. Bacterial cellulose synthesis (*bcs*) operons display highly variable genetic organizations across species[6] but they typically encode a c-di-GMP-regulated, membrane-embedded synthase/inner membrane transporter (BcsA). BcsA is usually accompanied by a co-catalytic membrane-anchored protein (BcsB), a periplasmic lyase (BcsZ), an outer membrane porin with periplasmic scaffolding motifs (BcsC), and additional structural and regulatory subunits, some of which are essential for secretion[6]. Although the structure and reaction cycle of the isolated BcsAB catalytic duo have been studied at nearly atomic resolution[8–10], almost no direct structural or functional studies have explored the role of accessory Bcs components.

Using the cellulose-producing *Escherichia coli* 1094 strain as a model, we analyzed the structure–function relationships between core and accessory Bcs subunits to provide a global view of the cellulose secretion machinery and pinpoint specific regulatory roles of individual components. Protein pulldown experiments show that most of the inner membrane and cytosolic Bcs subunits interact to form a multi-subunit protein macrocomplex. Electron microscopy reconstructions reveal a megadalton-sized assembly, in which multiple copies of the catalytic BcsAB duo arrange in an atypical asymmetric architecture and interact stably with cytosolic protein partners. Recombinant reconstitution of the Bcs macro-complex, structure–function analyses of several subcomplexes, and binary interaction screening via a two-hybrid approach point towards a multi-component cooperative system. Here we show that regulatory Bcs components contribute to secretion by affecting both the initial assembly and subsequent stability of the system and provide additional inputs for function regulation by the activating second messenger c-di-GMP. On the basis of these data, we propose functional models for nanoarray-like secretion of cellulose microfibers that would provide increased strength and biofilm forming capacity.

## Results

**Identification of Bcs components essential for secretion.** *E. coli*-like *bcs* operons are widespread among β-Proteobacteria and γ-Proteobacteria, including pathogenic and biocontrol organisms such as *Salmonella enterica* serovar Typhimurium, *Klebsiella pneumoniae*, *Burkholderia mallei*, *Shigella boydii*, *Yersinia enterocolitica*, *Vibrio fischeri*, and *Pseudomonas putida*[6] (Fig. 1a). Apart from the prevalent BcsABZC components, these typically express a MinD/ParA/Soj-homologous protein (BcsQ), a membrane-tethered protein with a periplasmic alkaline phosphatase-like domain (BcsG), a cytosolic c-di-GMP-binding protein with a GIL (GGDEF I-site like) domain (BcsE) and two small proteins (BcsR and BcsF)[6] (Fig. 1a, b; Supplementary Fig. 1a). Of these, BcsQ has been shown to be essential for cellulose biogenesis in vivo and to localize at the polar site for cellulose secretion and cell-to-cell adhesion[11], BcsE is known to be necessary for maximal cellulose production[12] and we show here, by introducing non-polar deletions, that wild-type levels of BcsG and BcsR are also essential for cellulose secretion (Supplementary Fig. 1b, c).

**Bcs components organize in a multi-component macro-complex.** To further investigate the central role of BcsQ in cellulose biogenesis, we undertook to determine whether the protein

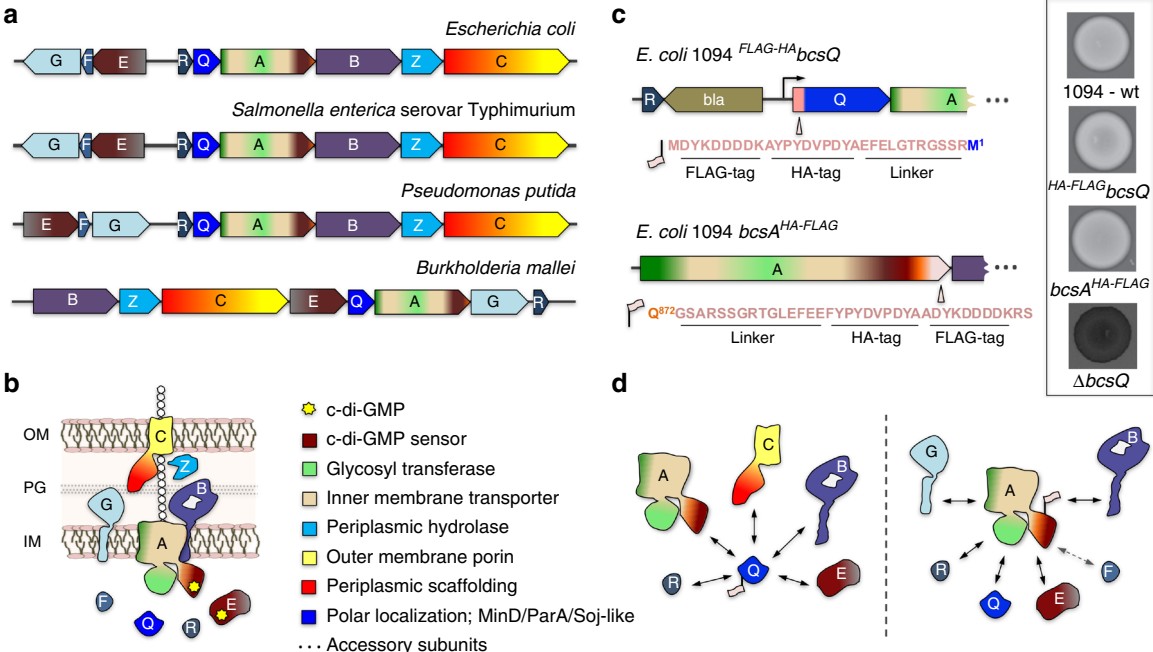

**Fig. 1** *E. coli*-like cellulose secretion systems—components and macrocomplex detection. **a** *E. coli bcs* operon organization (top) and examples of organisms sharing *E. coli*-like cellulose secretion system conservation. **b** Predicted localization and function of the encoded Bcs components. Domain architectures for all Bcs proteins are detailed in Supplementary Fig. 1. **c** Chromosomal epitope-tagging of the BcsQ and BcsA subunits and functional validation of the recombinant proteins. The sequence of the respective N-terminal and C-terminal epitope tags is shown on the left; calcfluor-binding and fluorescence of the recombinant *E. coli* 1094 strains is shown on the right. **d** Thumbnail summary of co-purifying Bcs partners as determined by anti-FLAG affinity purification and mass-spectrometry analyses

physically interacts with other Bcs subunits. To this end, we generated an *E. coli* 1094-derived strain, which expresses a functional FLAG-HA-tagged BcsQ and produces cellulose at wild-type levels (Fig. 1c). Cell fractionation experiments showed that overexpressed $^{FLAG-HA}$BcsQ partitions in both the soluble and membrane fractions and that its membrane association is fairly robust as washes with elevated salt, high pH, mild detergent, or 6 M urea solutions failed to fully extract it (Supplementary Fig. 2a). Interestingly, solubilization of the membrane fraction, followed by FLAG tag-dependent immunoprecipitation and mass-spectrometry analyses, detected BcsQ-dependent co-purification of multiple other Bcs components (Fig. 1d;

Supplementary Fig. 2b). Importantly, most of these interactions were confirmed by parallel affinity purification experiments on a strain expressing a functional BcsA$^{HA-FLAG}$ synthase from its native chromosomal locus (Supplementary Fig. 2b), as well as orthogonal bacterial two-hybrid experiments (see below), indicating the likely formation of a multi-component Bcs assembly.

**Electron microscopy reveals a large asymmetric Bcs assembly.** To gain structural insights into this macromolecular complex, we overexpressed the two *bcs* operons by inserting a two-directional constitutive-promoter cassette[13] at the *bcs* inter-operon region in the 1094 *bcsA*$^{HA-FLAG}$ genetic background (2K7 construct, Supplementary Fig. 3a–c). The purified membrane fraction was subjected to solubilization and anti-FLAG affinity purification, followed by glycerol-gradient centrifugation. Analysis of the gradient fractions by negative-stain electron microscopy (NS-EM), SDS-polyacrylamide gel electrophoresis (SDS-PAGE), liquid chromatography-coupled mass-spectrometry (LC-MS), and western blot (WB) reproducibly showed the formation of a stable complex consisting of most of the inner membrane-embedded (BcsA, BcsB, BcsF, and traces of BcsG) and predicted cytosolic Bcs components (BcsE, BcsQ, and BcsR) (Fig. 2a; Supplementary Figs. 2c and 3).

To determine the three-dimensional architecture of the Bcs macrocomplex, we proceeded to further improve the homogeneity of the sample by introducing mild, intramolecular chemical crosslinking. In particular, we applied the GraFix (gradient fixation) technique to covalently stabilize the complex during purification by density gradient ultracentrifugation, thus ensuring the purification of a homogeneous fraction separated from partially disassembled or aggregated complexes[14]. We collected an NS-EM dataset and obtained a refined, non-symmetrized structure reconstruction at 16.7 Å resolution (Fig. 2b–d; Supplementary Fig. 4). Volumetric analysis of the map indicates a molecular weight in the megadalton range that points towards the inclusion of multiple copies of individual Bcs components (Fig. 2d). Although the relatively low resolution did not allow us to determine the absolute handedness of the structure, all initial models refined to a layered, seashell-like architecture that spans about 215 × 200 × 150 Å in the three dimensions and is markedly different from that of other inner membrane transporters, bacterial secretory assemblies or EM and crystallographic reconstructions of the BcsAB catalytic duo from different organisms[8–10, 15, 16] (Fig. 2b–d; Supplementary Fig. 4). Although there is no intrinsic symmetry that can be assigned to

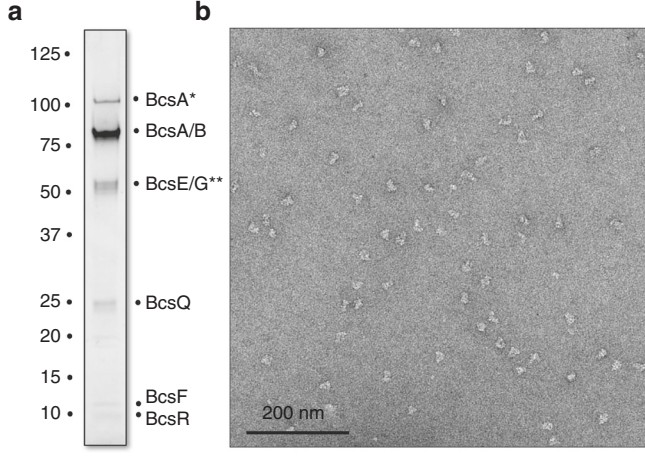

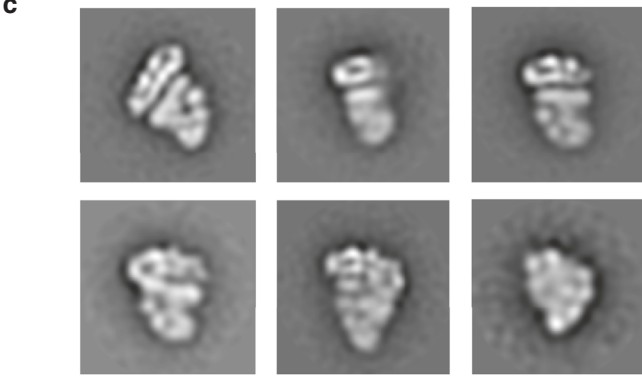

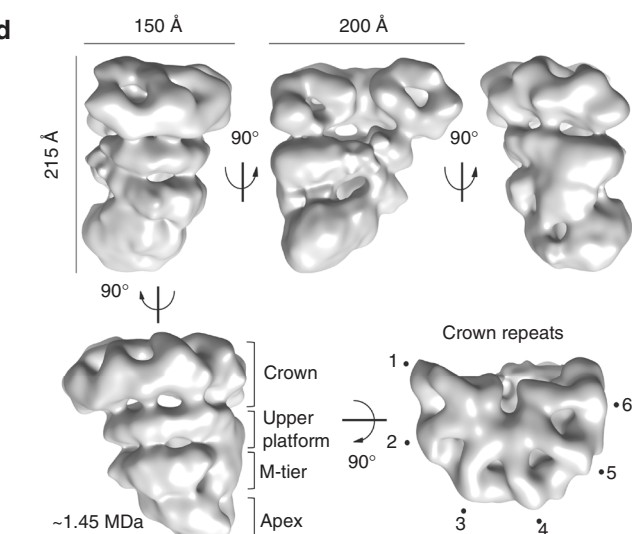

**Fig. 2** Electron microscopy analysis of the Bcs macrocomplex. **a** Purification of the Bcs macrocomplex expressed in the overexpression *E. coli* 1094 *bcsA*$^{HA-FLAG}$ 2K7 strain. SDS-PAGE of the affinity-gradient and density gradient-purified Bcs macrocomplex stained with Coomassie blue. Band labels are as identified by mass-spectrometry and immunoblotting. Asterisks denote presence of consistently identified contaminants discussed in Supplementary Fig. 2. **b** A representative micrograph of the negatively stained Bcs macrocomplex from the dataset used for image processing. **c** Representative views (class averages of 2D projections; not to scale) of the negatively stained Bcs macrocomplex. **d** Structure of the Bcs macrocomplex at 16.7 Å resolution. Different views and the relative rotational angles are shown. The characteristic layers and crown repeats are indicated in the bottom. The overall dimensions of the complex are indicated by size bars (top left and middle). The approximate molecular weight of the complex calculated from volumetric analyses of the reconstruction is indicated in megadaltons (MDa). Owing to the nature of the negatively stained sample, this estimation accounts only for the low-resolution envelope reconstruction and not intrinsic to the proteins electron density

the structure, its widest layer, to which we will refer to as the "crown", resembles a U-shaped radial arrangement of six repeated ring-shaped densities connecting through individual linkers to the upper platform of the assembly (Fig. 2d; Supplementary Fig. 5). These ring densities are reminiscent of the crystal structures of the N-terminal periplasmic domain of the accessory protein BcsB from *Rhodobacter sphaeroides* (Fig. 3a), which is predicted to be structurally conserved in *E. coli*[17] and has been shown to protrude about 60 Å into the periplasm to guide the newly synthesized cellulose chain exiting the BcsA glucan channel[8–10].

**Heterologous reconstitution of the Bcs macrocomplex**. As analyses of the crystallographic packing of the BcsAB[R.sphaeroides] duo in the deposited crystal structures did not identify similar BcsAB assemblies or any BcsB oligomerization interfaces that are likely to have biological significance (Supplementary Fig. 5a, b), we proceeded to corroborate the orientation of the complex relative to the bacterial membrane and the spatial arrangement of its key subunits. To this end, we sought to establish an IPTG-inducible, heterologous overexpression system where the macrocomplex as well as various subcomplexes were reconstituted in *E. coli* BL21 (DE3) cells. It is important to note that although the BL21 (DE3) strain contains a full set of conserved *bcs* genes, the cells do not secrete cellulose in any of the conditions tested, likely due to additional regulatory inputs (Supplementary Fig. 3b).

We cloned the *bcsRQA*[HA-FLAG]*B*, *bcsZC*, and *bcsEFG* regions in various combinations and under different promoters in the pCDF-Duet and pRSF-Duet co-expression-compatible vectors and introduced additional epitope tags for biochemical subunit identification (Fig. 3b; Supplementary Fig. 3). Less stringent affinity pulldowns did not show any significant difference in the purifications profiles with and without *bcsZC* components, which confirmed the absence of these periplasmic/outer membrane components in the Bcs assembly characterized above (Supplementary Fig. 3d). We continued by fully purifying the macromolecular complex expressed from the *bcs*[His]*RQA*[HA-FLAG]*B*- and *bcsEFG*-containing co-expression vector combination, determined its biochemical composition by a combination of SDS-PAGE, LC-MS, and western blot analyses (Supplementary Fig. 3e, f) and collected an NS-EM dataset for structure determination (Supplementary Fig. 4b). Characteristic views of the complex (class averages), as well as the three-dimensional structure reconstruction, revealed that the recombinantly expressed complex features virtually the same architecture as the Bcs macrocomplex expressed from the 1094 chromosomal locus (Fig. 3c). In addition, the purified complex was not only structurally intact but also displayed robust de novo cellulose synthetic activity in vitro (Fig. 3c), further validating Bcs macrocomplex assembly in the recombinant expression system.

**Subcomplex studies reveal system topology and cooperativity**. To determine whether the crown repeats indeed correspond to BcsB, we analyzed the protein's relative abundance in the purified complex, as well as the structure of several separate subcomplexes. SDS-PAGE analysis, where the intense band for the protein was identified by its redox-dependent mobility shift[18], LC-MS and WB on a tagged BcsB variant (Supplementary Fig. 3e, f), show that the protein indeed represents a major component of the Bcs macrocomplex. We next expressed and purified the *bcs*[His]*RQA*[HA-FLAG]*B*-pCDF-Duet construct separately and analyzed the glycerol-gradient fractions by biochemical assays and NS-EM (Fig. 3d; Supplementary Figs. 4–6). Reference-free class averages and volume reconstruction showed the formation of a relatively large inner membrane (IM) subcomplex

reminiscent of the crown and upper platform of the Bcs macrocomplex (Fig. 3d, e; Supplementary Fig. 5). Although some characteristic views showed incomplete assembly that can account for small conformational differences (Supplementary Fig. 5c, d), the majority of the particles showed the typical six repeats of the crown, whose large structural cavities are unlikely to correspond to the other major component of the subcomplex —BcsA. These data are also indicative that BcsG, predicted to fold into a periplasmic domain comparable in size to the BcsB N-terminal module[17] (Supplementary Fig. 5e), does not partake stoicheometrically in the resolved Bcs macrocomplex. On the basis of exhaustive BcsAB structural studies[8–10, 16], we can further attribute the upper platform to the inner membrane segments of partner BcsA copies (Supplementary Fig. 5g–j) but, interestingly, there is only a small region of additional density (the "stump") that cannot account for all of the BcsA cytosolic domains or additional subunits (Fig. 3d; Supplementary Fig. 5f, g). SDS-PAGE and WB analysis of the subcomplex further showed that BcsA undergoes reproducibly limited proteolysis and that the cytosolic subunits are present in substoicheometric ratio or non-detectable (Fig. 3d). As the "stump" density is observed at the level of 2D class averages and consequently the 3D reconstruction, we are inclined to concur that it likely does not correspond to a specific module but rather to residual density of the cytosolic BcsA and possibly BcsR/Q domains after averaging-out conformational heterogeneity. Taken together, these data strongly suggest that, whereas components encoded by the accessory *bscEFG* operon do not determine the oligomerization of the system per se, they contribute to cellulose biogenesis by regulating the stability of the multimeric synthase complex and, possibly, by an additional c-di-GMP regulatory input through BcsE[12].

Although BcsB is almost invariably found as an intra-operon partner of BcsA across bacterial genomes and is indispensable for BcsA synthase activity in *E. coli*[6, 18] (Fig. 1a; Supplementary Fig. 1b), it has been previously shown that only the membrane-associated C-terminal domain of BcsB suffices for synthase activity in vitro[18]. We therefore characterized the assembly of the Bcs macrocomplex in the context of N-terminal BcsB deletion (*bcs*[His]*RQA*[HA-FLAG]*B*[ΔNTD]-pCDF-Duet + *bcsEFG*-pRSF-Duet) as an orthogonal approach to validate the attribution of crown densities to BcsB (Supplementary Figs. 3e and 6a–c). Although the sample was characterized by increased heterogeneity and instability, we observed characteristic class averages consistent with the inner membrane and cytosolic densities as deduced above (Supplementary Fig. 6c). Taken together, these data are indicative that Bcs macrocomplex assembly and BcsAB oligomerization are likely stabilized but not primed by BcsB crown assembly and that the regulatory cytosolic components assemble stably with the catalytic moieties.

As a sizable cytosolic component, c-di-GMP-sensing BcsE is expected to be a major contributor to the apical volumes of the Bcs macrocomplex structure and we examined whether it can assemble into the Bcs macrocomplex in the absence of its intra-operon partners. Importantly, a complex co-expressed from the *bcs*[His]*RQA*[HA-FLAG]*B*-pCDF-Duet + *bcsE*-pRSF-Duet construct combination showed severely compromised BcsE association and the characteristic views and biochemical profile of the purified complex were predominantly similar to those of the inner-membrane subcomplex lacking BcsE co-expression (Supplementary Fig. 6d–h). This, together with the BcsE–BcsF interactions captured by bacterial two-hybrid experiments (below) suggest, yet again, cooperative assembly among Bcs subunits.

Finally, considering the central role of BcsQ in cellulose secretion, we wanted to evaluate Bcs macrocomplex assembly in the context of a Δ*bcsQ* deletion and an inactivating BcsQ-G[14]S

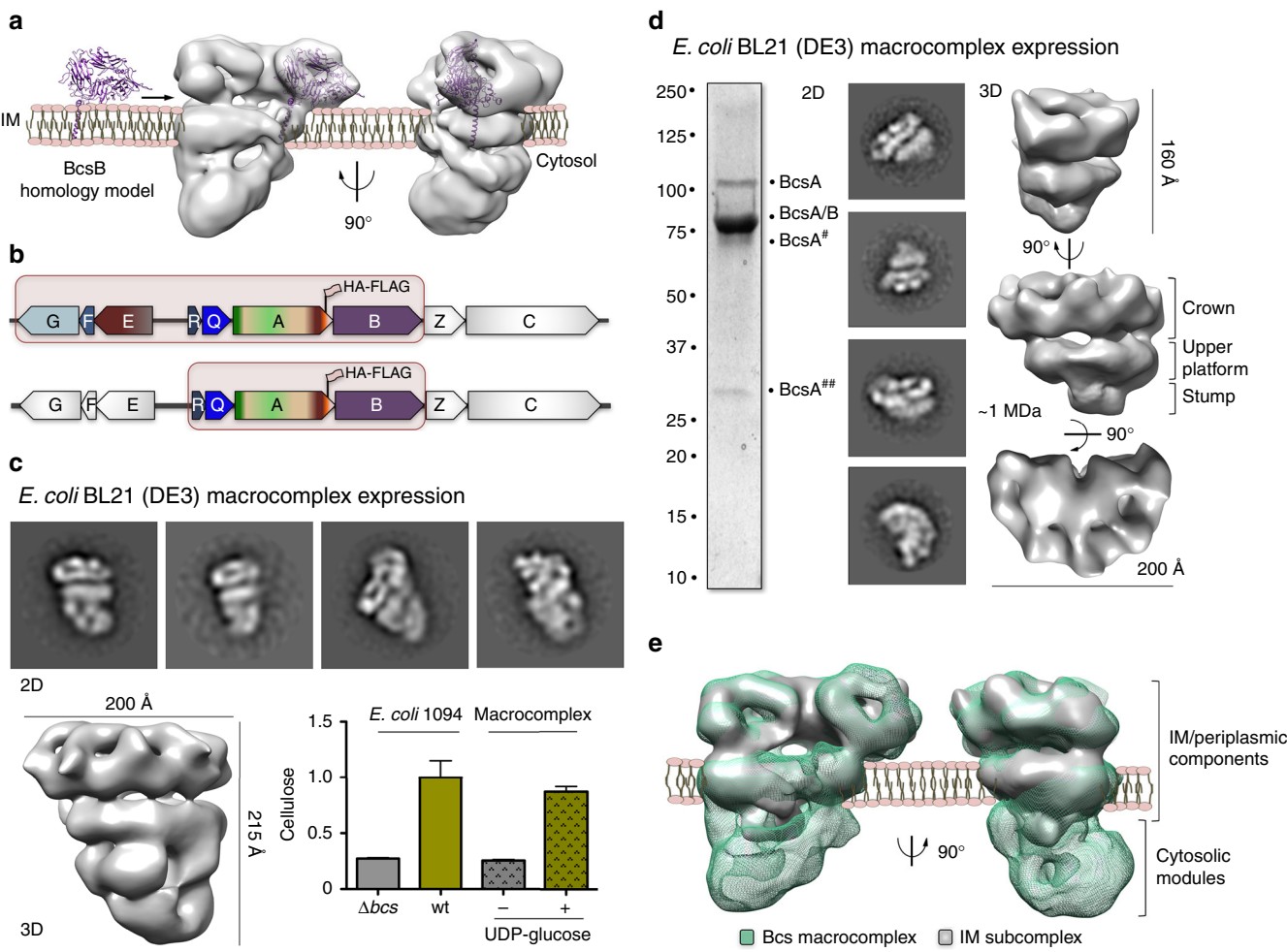

**Fig. 3** Map interpretation and electron microscopy studies of an inner membrane (IM) subcomplex. **a** Representative fitting of a BcsB homology model in a crown repeat volume and proposed orientation of the Bcs macrocomplex relative to the inner membrane. **b** Co-expressed subunits for recombinant overexpression and Bcs complex reconstitution in BL21 (DE3) cells (macrocomplex: top; IM subcomplex: bottom). **c** Structural and functional analyses of the recombinantly expressed Bcs macrocomplex. Representative 2D class averages are shown on the top (not to scale); the 3D structure reconstruction is shown on the bottom. End-point de novo cellulose synthesis by the purified complex is shown on the bottom; results are representative of two biological replicates. The data is plotted relative to cellulose detection in a wild-type E. coli 1094 cell culture, with bcs operon-deleted strain serving as negative control. **d** Structural analyses of the IM subcomplex. A coomassie-stained SDS-PAGE gel, 2D class averages (not to scale) and different views of the 3D structure reconstruction are shown from left to right. The corresponding molecular weight is indicated in megadaltons and the dimensions of the subcomplex are indicated by size bars. **e**. Fitting of the IM subcomplex into the Bcs macrocomplex volume and proposed orientation relative to the inner membrane

point mutation[11] (Supplementary Figs. 1b and 7). Surprisingly, dot-blot assays showed that both variants, and especially the deletion mutant, were severely compromised in expression of partner Bcs subunits (Supplementary Fig. 7a). Macrocomplex assembly was severely disrupted in the $\Delta bcsQ$ background and featured a non-characteristic biochemical profile (Supplementary Fig. 7c). Even more surprisingly, residually co-expressed Bcs components showed catalytic activity in vitro (Supplementary Fig. 7d), suggesting that the deleterious effects of the mutations occur at the level of Bcs subunit expression, folding and complex assembly rather than abolishing BcsAB-specific synthase activity per se. This underscores further the role of accessory subunits in regulating cellulose biogenesis by affecting both the expression and stability of the system, as well as the need for holistic structure–function analyses beyond the catalytic BcsAB pair.

**Binary Bcs interaction studies reveal function determinants**. As the current structural model of the Bcs macrocomplex does not allow us to fully resolve the electron densities and structure of

individual subunits, we resorted to a bacterial two-hybrid assay[19] to identify direct interactions among inner membrane and cytosolic Bcs full-length components and/or functionally important structural motifs (Supplementary Fig. 8). The data uncover a complex interaction network that provides further insights into direct binary interactions beyond the well-studied BcsAB catalytic duo (Fig. 4a). Importantly, we observe that BcsQ, a protein essential for cellulose production, oligomerizes and interacts with several other Bcs components. These include the catalytic glycosyl transferase (GT) domain and the activating c-di-GMP sensing PilZ module of BcsA, BcsE, and its c-di-GMP-sensing GIL domain, and the small protein BcsR, also essential for cellulose production (Supplementary Fig. 8). These data suggest that the BcsQ protein likely occupies central, M-tier density in the macrocomplex and could directly affect BcsA stability, c-di-GMP-dependent activation, and processive catalysis within the assembled secretion system. Interestingly, BcsQ binary interactions are disrupted in the cellulose-deficient BcsQ-G$^{14}$S mutant (Supplementary Fig. 8), which can explain why, although catalytic

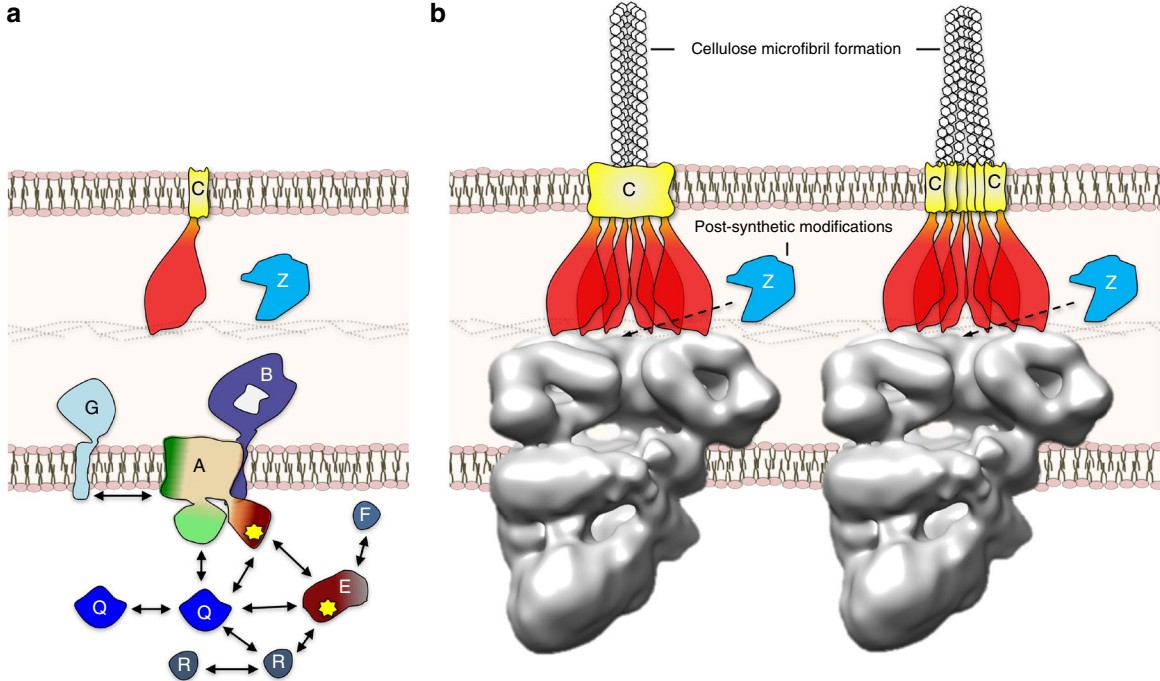

**Fig. 4** Bcs protein interactions and functional model of cellulose microfibril secretion. **a** A summary of the direct protein–protein interactions as observed by bacterial two-hybrid screen. **b** Proposed microfibril formation at the periplasmic or cell surface level. The hexameric assembly is proposed to secure a cell pole-localized nanoarray for cellulose biogenesis and microfibril formation, the open architecture is proposed to secure access for the hydrolizing endo-1,4-β-D-glucanase BcsZ necessary for maximal cellulose production. The BcsC TPR-rich periplasmic motifs are shown in red, whereas the C-terminal outer membrane porin domain is shown in yellow. Microfibril formation at the periplasmic level would require the assembly of a wider composite porin for outer membrane secretion (left)

activity, complex assembly and polar BcsQ localization are preserved to some extent in the BcsQ-G[14]S background[11] (Supplementary Fig. 7c, d), the intact NTPase domain of the protein remains essential for the full assembly and function of the cellulose system in vivo (Supplementary Fig. 1b).

Furthermore, we observe specific interactions of other Bcs components with functionally important BcsA structural motifs[20]. (Supplementary Fig. 8c–f). These results suggest a more complex regulation of the system at multiple levels. The direct interactions between c-di-GMP sensing modules (BcsE[GIL] and BcsA[PilZ]), as well as their binary interactions with the essential for the system's assembly and function BcsQ, suggest that c-di-GMP binding can stabilize the assembled secretion system similarly to the dinucleotide's role in the secretion of another prevalent, synthase-dependent exopolysaccharide of Gram-negative species, poly-N-acetyl-glucosamine (PNAG)[21] (Supplementary Fig. 9a). In line with this, we observe that the BcsE[GIL]-BcsQ interaction is dependent on the integrity of the c-di-GMP binding I-site of BcsE (R[415]xxD), whereas BcsQ interacts stably with BcsA[PilZ] (Supplementary Fig. 8e). The latter interaction was not disrupted even in the context of mutations aimed to disrupt the c-di-GMP-binding PilZ domain linker (R[696]xxxR), indicating that this regulatory site is likely exposed for c-di-GMP recognition, at least in the context of this binary complex. Furthermore, the interaction between BcsE and BcsA[PilZ] requires the integrity of the c-di-GMP-coordinating motifs of both modules (Supplementary Fig. 8f). As the dinucleotide has been shown to bind as an intercalated dimer to both the BcsA[PilZ] domain and to the RxxD I-site of active and degenerate diguanylate cyclases, a stabilizing role for c-di-GMP at the interface of the different Bcs subunits is not unlikely. Conversely, BcsE oligomerization (Supplementary Fig. 8a, b) and c-di-GMP binding can further provide c-di-GMP protection and increase the local concentration of the dinucleotide, to contribute

to the full activation of the system through direct dinucleotide signal relay. The observation that E. coli BL21 (DE3) cells fail to secrete cellulose even after complementation with the full 1094 bcs operon (Supplementary Fig. 3b), indicates signaling specificity that could indeed involve additional upstream or downstream factors and require an isolated rather than generic c-di-GMP activating signal, which remains to be further examined.

The two-hybrid data demonstrates additional binary interactions between multiple other E. coli-like specific modules. For example, BcsE was shown to interact directly with BcsF (Supplementary Fig. 8a, b), which can explain its disrupted association with the complex in the absence of its intra-operon partners (Supplementary Fig. 6d–h). Interactions between the BcsA[NTD], an N-terminal domain found in E. coli-like synthases but absent in the best-studied R. sphaeroides homolog, and the predominantly periplasmic and essential BcsG, on the other hand, suggest secretion system subclass idiosyncrasies that might govern further system assembly and interactions with the periplasmic and outer membrane export components.

## Discussion

We report here detailed structural and assembly analyses of the cellulose biosynthetic (Bcs) machinery in E. coli, using the 1094 strain as an excellent model for the exopolysaccharide's role in environmental persistence and biofilm formation by multiple evolutionary-related species.

The structural and interaction data shown here demonstrate that cellulose-secreting E. coli-like bacteria have evolved a complex multi-component secretion system, whose inner membrane and cytosolic components assemble into a stable macrocomplex with previously unobserved asymmetric architecture. This macrocomplex is comparable in size to other studied systems for the secretion of hydrophilic biopolymers[15, 22, 23] (Supplementary

Fig. 9b) but encompasses only the inner membrane of the cell envelope. Although it remains to be determined how the processive export of polymeric cellulose is carried out through the periplasm and outer membrane of the cell, the multimeric assembly visualized here leads us to propose functional models for cellulose secretion.

Although some bacteria, such as *G. xylinus*, present an ordered array of cellulose synthase complexes on their envelope to secrete crystalline cellulose[24], *E. coli* BcsQ, and secreted cellulose have been shown to localize to the poles[11] and the product is proposed to be amorphous, non-bundled polymer similar to that of the best-studied system of *R. shpaeroides*[20, 25]. Recent work on the latter has shown, however, that artificial nanoarrays of the BcsAB catalytic complex leads to spontaneous cellulose microfibril formation[26]. It is therefore possible, that the hexameric architecture visualized here has evolved to provide a similar nanoarray for the secretion of stronger cellulose, where the individual polymers can be guided and secreted through individual porins to bundle up at the cell surface (Fig. 4b). Alternatively, individual cellulose polymers could be weaved at the periplasmic level and exported through a composite outer membrane β-barrel, as observed in other secretion systems (Fig. 4b)[15, 27]. Importantly, BcsC contains a stretch of ~10 amino acids that are likely to form a flexible linker (up to ~3.5 nm in extended form) at what is predicted to be the membrane-proximal interdomain boundary (Supplementary Fig. 1a), which can allow for significant conformational variability between BcsC subunits and, potentially, for an asymmetric packing as in the proposed model. Cellulose bundling is also consistent with earlier EM data where secreted cellulose fibers are visible at the cell surface even at low magnifications[11]. Finally, the open, non-cylindrical architecture observed here would also provide access for post-synthetic modifications by the periplasmic hydrolase BcsZ, which is necessary for maximal cellulose secretion and/or biofilm formation in vivo[6].

Biofilm cellulose provides a fitness and resistance advantage in many free-living and pathogenic species and is of increasing interest for diverse biotechnological applications[4, 6, 28]. Although many of the system's properties described here are governed by *E. coli*-like specific Bcs components, the functional conservation among synthase-dependent polysaccharide secretion systems (Supplementary Fig. 9a) begs the question whether similar macrocomplexes assemble in more distant species[5]. Given that these all secure the biosynthetic activities, physical conduit and energetics for the export of hydrophilic biopolymers through the complex bacterial envelope, we propose that they comprise a distinct type of secretory nanomachines (Supplementary Fig. 9b) whose intricate regulatory mechanics are only beginning to emerge.

## Methods

**Generation of *E. coli* chromosomal mutants and gene fusions**. Standard protocols were used for molecular cloning, PCR, DNA analysis, and transformation. All strains are available upon request. Deletions and insertions in the *E. coli* 1094 chromosome were generated by the λ-red linear DNA gene inactivation method using a three-step PCR procedure[29]. All mutations were checked by PCR and DNA sequencing. The 1094 Δ*bcsA*, Δ*bcsB*, Δ*bcsC*, Δ*bcsG*, Δ*bcsQ*, and Δ*bcsR*-mutant strains, displaying a defect of cellulose production, were complemented upon introduction of derivative of the low-copy spectinomycin-resistant plasmid pam238, expressing a corresponding wild-type allele of the *bcsA*, *bcsB*, *bcsC*, *bcsQ* *bcsR*, or *bcsG* genes. The 1094$^{\text{FLAG-HA}}$*bcsQ* and the 1094$^{\text{FLAG-HA}}$*bcsQ-G$^{14}$S E. coli* strains were constructed by inserting a genetic cassette corresponding to an ampicillin resistance gene followed by a constitutive λ promoter (PcL) to induce a FLAG-HA9 epitope tag coding tandem inserted in-frame with the 5′ end of the *bcsQ* gene in *E. coli* 1094 wild-type or *bcsQ-G$^{14}$S*, respectively[11]. The 1094 *bcsA*$^{\text{HA-FLAG}}$ strain was constructed by introducing a genetic cassette comprising a FLAG-HA9 epitope tag coding tandem inserted in-frame at the 3′ end of the *bcsA* gene, followed by a kanamycin resistance gene flanked by flippase recognition target (FRT) sites. The resistance gene was subsequently excised from the chromosome

using a pCP20 plasmid-encoded FRT flipase (Flp)[30]. A similar procedure was used for the construction of a 1094 *bcsB*$^{\text{HA-FLAG}}$ strain as well.

For the construction of the *bcs* overexpression strain for structural studies, a two-directional inducible-promoter cassette[13] was introduced by a two-step procedure in the *bcs* inter-operon region in wild-type *E. coli* 1094 and was subsequently transferred in the 1094 *bcsA*$^{\text{HA-FLAG}}$ background to yield the 1094 *bcsA*$^{\text{HA-FLAG}}$ 2K7 overexpression strain. Briefly, in the first step a *km*RExTET cassette was introduced in front of *bcsR*[13]. In a second step, the region of the cassette upstream of the Ptet promoter was substituted by an ampicillin resistance gene and a constitutive λ promoter (PcL). In the resultant strains, the *bcsRQABZC* operon is constitutively expressed under a derepressed Ptet promoter and the *bcsEFG* operon is constitutively expressed under the constitutive PcL promoter. The expression of cellulose and Bcs protein expression were evaluated by calcofluor-binding and dot-blot assays (below).

**Calcofluor-binding cellulose secretion assay**. *E. coli* 1094 wild-type and derivative strains were grown at 37 °C in liquid LB medium supplemented with appropriate antibiotics, when needed. Five microliters of overnight culture were spotted onto LB-CF plates containing LB-agar, supplemented 0.02% calcofluor (Sigma), and 1 mM HEPES pH 7–7.6. The spotted drops were allowed to dry and plates were incubated for 24 h at 30 °C. Cellulose secretion was evaluated by the binding of calcofluor, as visualized by colony fluorescence under ultraviolet light. Pictures were taken using a G:BOX imaging system from Syngene.

**BcsQ cell localization and membrane washing experiments**. Exponential phase 1094$^{\text{FLAG-HA}}$*bcsQ* cells grown in M63B1 medium with 0.4% glucose were pelleted by centrifugation (5000×*g*, 20 min, 4 °C), washed in PBS and mechanically disrupted using a French press. After removal of the non-lyzed cells by centrifugation, the membrane fraction was pelleted by ultracentrifugation for 1 h at 165,000×*g* and 4 °C. The membranes were then washed with PBS and resuspended in ice-cold PBS, 500 mM KCl, 100 mM sodium carbonate pH 11, 0.1% Triton X-100, or 6 M urea. After incubation for 30 min, the samples were subjected to a second ultracentrifugation step as above. BcsQ distribution in the membrane vs. soluble fractions was analyzed after each ultracentrifugation by SDS-PAGE and immunoblot.

**Initial affinity purification of $^{\text{FLAG-HA}}$BcsQ and BcsA$^{\text{HA-FLAG}}$**. Exponential phase *E. coli* 1094$^{\text{FLAG-HA}}$*bcsQ* bacteria were washed and resuspended in buffer A containing 20 mM Tris pH 7.5, 150 mM NaCl, 10% glycerol, and 0.2 mM EDTA. After removal of the non-lyzed cells by centrifugation, the soluble fraction was collected as supernatant by ultracentrifugation (165,000×*g*, 1 h, 4 °C). The pelleted membrane fraction was washed with buffer A and resuspended in solubilization buffer containing all buffer A components and 0.1% Triton X-100. After 30 min incubation on ice, the solubilized membrane fraction was collected as supernatant by ultracentrifugation at (165,000×*g*, 1 h, 4 °C). The latter was incubated with Anti-FLAG M2 resin (Sigma-Aldrich) for 2 h at 4 °C under mild agitation. After extensive washes with buffer A, the affinity column was eluted with buffer A supplemented with FLAG peptide (Sigma-Aldrich), according to the manufacturer's instructions.

**Mass-spectrometry identification of binding partners**. Mass-spectrometry analyses were carried out by the Proteomics Platform at the Institut Pasteur as previously published[31]. For each in-solution sample, 10 μg of total protein were dried and resuspended in 100 mM Tris-HCl pH 8.5, supplemented with 8 M urea. Sample reduction, acetamidation, and tryptic digestion were done using standard protocols[32]. Digests were analyzed on an LTQ-Orbitrap Velos instrument (Thermo Fisher Scientific) equipped with an Ultimate 3000 nano-HPLC system (Dionex). Samples were desalted and loaded on a C18 trap column (Dionex) before being further separated on a C18 PepMap100 column (Dionex) under a linear gradient of acetonitrile. The instrument method for the LTQ-Orbitrap Velos was set up in the data dependent acquisition mode. After a survey scan at resolution of 60,000, the 20 most intense precursor ions were selected for CID fragmentation in the ion trap. Minimum signal threshold for triggering an MS/MS event was set to 5000 counts. For internal mass calibration the 455,120025 ion was used as lock mass. Charge-state screening was enabled, and precursors with unknown charge state or a charge state of 1 were excluded. Dynamic exclusion was enabled for 90 s.

Raw files were processed with Mascot v.2.2 (Matrix Science) as search engine on Proteome Discoverer v.1.2 (Thermo Fisher Scientific) against the SwissProt database (all entries and reversed sequences or 1,044,038 entries in total). Trypsin was chosen as specific enzyme with a maximum number of two missed cleavages. Possible modifications included carbamidomethylation of cysteines (fixed), oxidation of methionines (variable), and formation of pyroglutamate (on N-terminal Glu, variable). Mass tolerance for MS was set to 10 ppm and 0.5 Da was used for MS/MS. Probability assignment and validation were performed using the Scaffold software v.3.5.1 (Proteome Software Inc.)[33]. A false-discovery rate of 1% was used for both peptide and protein identification. Reverse and contaminant proteins were excluded and only proteins identified with a minimum of two peptides were considered. To underscore interactions among Bcs components, only detected Bcs proteins were presented in Fig. 1d and Supplementary Fig. 2b.

Protein identification by mass-spectrometry was performed on Coomassie-stained and excised SDS-PAGE gel bands following Bcs complex overexpression and purification for structural studies (see below). The protein treatment and peptide detection protocol was the same as above but with standard in-gel tryptic digestion. Raw files were processed with MaxQuant 1.4.1.2 as search engine[34] and against a concatenated database containing all *E. coli* BL21 (DE3) proteins, the *E. coli* 1094 Bcs proteins (including epitope modifications as expressed), as well as the typical contaminants. Results interpretation was subject to the same criteria as above. Consistently identified *E. coli*-contaminant proteins with relevant molecular weights were underscored in the data representation.

**Recombinant overexpression**. Using genomic DNA from the 1094 *bcsA*[HA-FLAG] strain as a template, the *bcsRQA*[HA-FLAG]*B*, *bcsEFG*, and *bcsZC* regions were PCR-amplified with appropriate restriction sites introduced in the 5′ primer overhangs (sense/antisense *Pst*I/*Not*I, *Bam*HI/*Not*I, and *Hind*III/*Xho*I, respectively). In parallel, the pCDF-Duet and pRSF-Duet vectors were also PCR-amplified to include the respective restriction sites for in-frame ligation under pCDF-Duet Promoter 1, pRSF-Duet Promoter 1, and pRSF Promoter 2, respectively. All PCR products were subsequently digested with the respective restriction enzyme pair (New England Biolabs), gel-purified, ligated using T4 DNA ligase (New England Biolabs), transformed into chemically competent DH5α cells, and plated on LB-agar plates containing an appropriate antibiotic (100 μg ml⁻¹ streptomycin and 50 μg ml⁻¹ kanamycin for the pCDF-Duet and the pRSF-Duet constructs, respectively). Single colonies were grown in 5 ml liquid LB medium at 37 °C overnight and the plasmid DNA was extracted using NucleoSpin® Plasmid preparation kit according to the manufacturer's instructions (Macherey-Nagel). Positive clones were identified by restriction digestion and DNA sequencing. Although an octahistidine tag was introduced to BcsR upon the initial cloning step, additional Strep II-tags were introduced to BcsE or BcsQ using inverse PCR. The sequence of all modified proteins is shown in the data figures. Inverse PCR was further used for in-frame deletions of *bcsQ*, *bcsFG*, and the N-terminal periplasmic domain of *bcsB* (*bcsB*[ΔNTD]), as well as for the construction of the BcsQ[G14S] mutant. For co-expression, chemically competent BL21 (DE3) cells were co-transformed with the purified plasmids and plated on LB-agar plates with antibiotic concentrations reduced to two-thirds of the ones stated above. After overnight incubation of the plates, multiple colonies of the transformed BL21 (DE3) cells were picked and grown at 37 °C in antibiotics-supplemented TB medium to optical density at 600 nanometers (OD₆₀₀) of 0.8–1.2, upon which the cultures were transferred to 17 °C and induced with 0.7 mM isopropyl-β-D-thiogalactopyranoside (IPTG, Eurobio) for 16 h.

**Complex purification for structural studies**. For complex purification, 1094-derived strains were grown in M63B1 medium at 37 °C and BL21 (DE3) cells in TB at 37 °C/17 °C (growth/induction) in the presence of appropriate antibiotics. Cells were pelleted by centrifugation (5000×*g*, 20 min, 4 °C) and the pellets were resuspended in ice-cold buffer B containing 20 mM HEPES pH 8.0, 120 mM NaCl, 10% glycerol, 5 mM MgCl₂, 10 μM adenosine-5′-[(β,γ)-methyleno]triphosphate (AppCp, Jena Bioscience), 2 μM cyclic diguanylate monophosphate (c-di-GMP, Sigma-Aldrich), 250 μM cellobiose, 0.5 mg ml⁻¹ *Aspergillus niger* cellulase (Sigma-Aldrich), 100 μg ml⁻¹ lysozyme, and 1 tablet per 50 ml complete EDTA-free protease inhibitors (Roche). The cells were subsequently disrupted using an Emulsiflex-C5 high-pressure homogenizer (Avestin) and the lysates were pre-cleared by a low-speed centrifugation step (12,000×*g*, 15 min, 4 °C). Membranes were pelleted by high-speed centrifugation using an SW 32 Ti Beckman rotor (29,500 rpm, or 65,000–148,000×*g*, for 1 h at 4 °C) and resuspended in solubilization buffer containing all buffer B components but lysozyme and cellulase, as well as a mix of detergents at the following final concentrations: 0.4% w/v digitonin (Sigma-Aldrich), 0.4% w/v *n*-dodecyl-β-D-maltopyranoside (anagrade β-DDM, Anatrace), 0.4% w/v decyl maltose neopentyl glycol (DM-NPG, Anatrace), and 0.2% w/v lauryl maltose neopentyl glycol (LM-NPG, Anatrace). The detergent mix was empirically optimized using published studies on other secretion system assemblies as a starting point[22, 35]. After incubation for 90 min at 22 °C and under mild agitation, the solubilized membrane fraction was cleared by a second high-speed centrifugation step as above. The supernatant was then incubated with ANTI-FLAG® M2 affinity gel (100 μl resin per litre of induced culture, Sigma-Aldrich), under mild agitation at 4 °C for 1 h. After gravity elution of the non-bound fraction, the resin was washed extensively (>30 column bed volumes) with affinity buffer containing all buffer B components but lysozyme and cellulase, as well as 0.008% w/v LM-NPG. The bound complexes were eluted using four column bed volumes of ellution buffer (affinity buffer supplemented with 3X FLAG® peptide at 100 μg ml⁻¹), concentrated on a 100 kDa cut-off Amicon® Ultra (MerckMillipore) centrifugal filter and loaded on a 10–40% glycerol density gradient containing all other components of the affinity buffer. For purification of the Bcs macrocomplex from the 1094 *bcsA*[HA-FLAG] 2K7 overexpression strain for EM data collection, 0.1% v/v of glyceraldehyde was added to the heavy fraction of the gradient prior to mixing[14]. The samples were then subjected to ultracentrifugation in a Beckman SW 55 Ti rotor at 36,000×*g* for 13 h at 4 °C and fractionated manually. Fractions were analyzed by electron microscopy for single-particle abundance and homogeneity, and EM data was collected on enriched

homogeneous fractions. Such fractions were also concentrated and analyzed by SDS-PAGE, western blots, and mass-spectroscopy.

**Electron microscopy and image processing**. Structural analyses of the various Bcs complexes was performed by negative-stain electron microscopy. Overall, 5 μl of density gradient fractions (concentrations ~0.01 μg μl⁻¹) were spotted on glow-discharged carbon-coated copper grids (Agar Scientific). After 1 min incubation, the extra liquid was blotted off and the grids were passed sequentially through three drops of 2% w/v uranyl acetate solution, with 30 s incubation in the last one before blotting and air-drying. All data was collected at the electron microscopy platforms of the Institut Pasteur (single-tilt) and the I2BC (random conical tilt reconstruction).

Data on the (i) Bcs macrocomplex purified from the 1094 *bcsA*[HA-FLAG] 2K7 strain, (ii) the macrocomplex purified from BL21 (DE3) cells upon *bcs*[His]*RQA*[HA-FLAG]*B*- and *bcsEFG*-containing vector co-expression, and (iii) the inner membrane subcomplex purified from BL21 (DE3) cells upon *bcs*[His]*RQA*[HA-FLAG]*B* construct expression were collected automatically using the EPU software on a Tecnai F20 FEG microscope, operating at a voltage of 200 kV and equipped with an FEI Falcon II direct electron detector (used nominal magnification 50,000, pixel size of 1.9 Å², dose rate 12 electrons per Å² s, exposure time 1 s).

For the Bcs macrocomplex purified from the 1094 *bcsA*[HA-FLAG] 2K7 strain (i), particles were picked automatically using EMAN2/BOXER[36], curated manually to remove aggregates and stain artifacts and extracted in boxes of 210 × 210 pixels. The defocus value was estimated and the contrast transfer function (CTF) was corrected by phase flipping using EMAN2 (e2ctf). A particle stack was generated to include a total of 44,056 particles with defocus values between 0.4 and 3.5 μm. All classification and refinement steps were performed using RELION 1.4[37]. Two rounds of reference-free class-averaging were used further clean up the automatically selected dataset upon which a final dataset of 24,769 particles was assembled. An initial 3D model restricted to 60 Å resolution was generated in EMAN2 (e2initialmodel) and was input for 3D classification in RELION 1.4 using 3 classes. As only minor conformational changes were observed, refinement of a single 3D model was performed on the ensemble of particles. RELION 1.4 post-process procedure was used to obtain a final reconstruction at 16.7 Å resolution, after masking and based on the "gold standard Fourier shell correlation criterion" (FSC > 0.143). Before visualization, the density map was sharpened by applying a negative ad hoc B-factor of −1000. 2D reprojections of the final 3D model were generated in EMAN2 using the e2project3d function and the results were matched to characteristic 2D averaged views by visual inspection.

For the macrocomplex purified from BL21 (DE3) cells upon *bcs*[His]*RQA*[HA-FLAG]*B*- and *bcsEFG*-containing vector co-expression (ii), particles were similarly picked, extracted, estimated for defocus and CTF-corrected in EMAN2 to assemble a set of 10,173 particles (210 × 210 pixels each) with defocus values of 0.4–3.5 μm. Three rounds of 2D reference-free class averaging in RELION 1.4 yielded a final set of 7229 particles. These were used to refine in RELION 1.4 a 3D initial model generated in EMAN2 to a final resolution of 26.1 Å, using the procedures and criteria as above. For the inner membrane subcomplex purified from BL21 (DE3) cells upon *bcs*[His]*RQA*[HA-FLAG]*B*-pCDF-Duet construct expression (iii), particles were similarly picked, extracted, estimated for defocus and CTF-corrected in EMAN2 to assemble a set of 127,430 particles (180 × 180 pixels each) with defocus values of 0.5–3.5 μm. Two rounds of reference-free class-averaging was used to clean up the data and a final set of 20182 particles was input for 3D classification in RELION 1.4 against an initial model generated in EMAN2. Two of the three classes that were generated displayed consistent conformation and the corresponding 14,362 particles were used for final 3D refinement and structure reconstruction at 21.1 Å resolution, using the procedures and criteria as above.

Electron micrographs of the subcomplex resulting from the *bcs*[His]*RQA*[HA-FLAG]*B*[ΔNTD]-pCDF-Duet & *bcsEFG*-pRSF-Duet co-expression (iv) were collected manually using an FEI Tecnai T12 BioTWIN LaB6 electron microscope operating at 120 kV at nominal magnification of 68,000 (pixel size 1.55 Å²) and 0.8–3 μm defocus. Images were recorded on a Gatan Ultrascan 4000 CCD camera. A total 40845 particles (260 × 260 pixels) were automatically picked, extracted, defocus-estimated, and CTF-corrected in EMAN2, and subjected to two rounds of reference-free 2D classification in RELION 1.4.

Finally, the complexes purified from BL21 (DE3) cells upon *bcs*[His]*RQA*[HA-FLAG]*B*-pCDF-Duet + *bcsE*-pRSF-Duet co-expression (v) and *bcs*[His]*RQA*[HA-FLAG]*B*-pCDF-Duet + *bcs*[Strep]*EFG*-pRSF-Duet co-expression (vi) were visualized at adjusted concentrations immediately after elution from the ANTI-FLAG® M2 affinity resin. Electron micrographs were collected manually at the Tecnai T12 microscope as above and particles were automatically picked, extracted, defocus-estimated and CTF-corrected in EMAN2 to yield a set of 16,877 particles (260 × 260 pixels) and 17,193 particles (260 × 260 pixels), respectively. These were subjected to a single round of 2D reference-free classification in RELION 1.4 each and the relative abundance of the fully assembled Bcs macrocomplex was estimated based on unambiguous characteristic views.

For random conical tilt reconstructions (rct), 3633 particles were imaged at 0° and 50° stage tilt in thicker negative-stain using a Tecnai T12 microscope operating at 100 kV and equipped with a K2 Base direct electron detector (nominal magnification 15,000, pixel size 2.52 Å²). Tilt pairs of particles were boxed and extracted and the untilted particles were defocus-estimated, CTF-corrected, and

subjected to 2D classification in EMAN2. The untilted and tilted datasets were subsequently low-pass-filtered to 25 Å to reduce noise artefacts and input together with the best 2D classes for random conical tilt reconstruction (e2rct), generating a 3D rct model per class. The resulting models were then restricted to 60 Å and fed as initial models for 3D classification and refinement of the 1094 $bcsA^{HA-FLAG}$ 2K7 dataset of 24,769 particles described above. Although all final models featured the same characteristic architectural features, the resolution of the raw data, initial models, and final refined reconstructions did not allow unambiguous assignment of absolute handedness to the structure.

**Structure display and analysis**. The three-dimensional structure reconstructions were displayed in UCSF Chimera[38] and segmentation of the volumes was performed with the SEGGER module implemented in the package. Volumetric analysis was also performed in UCSF Chimera[38] with average protein density of 0.81 Da per $Å^3$ used for the volume-to-mass conversion after adjusting the threshold based on assigning the crown densities to the BcsB periplasmic modules. Owing to the nature of the negatively stained sample, this estimation accounts only for the low-resolution envelope reconstruction and not intrinsic to the proteins electron density. Homology models generated in Robetta[39] and deposited crystal structures were displayed in UCSF Chimera[38] or the PyMOL molecular graphics systems (Schrödinger, LLC). UCSF Chimera[38] was used for automatic docking of the BcsB model in the crown volume after map fragmentation, as well as for fitting of the inner membrane subcomplex into the map of the Bcs macrocomplex.

**SDS-PAGE, western, and dot-blot assays**. SDS-PAGE was performed in Bio-Rad Mini-PROTEAN systems using 4–20% TGX precast gradient gels and standard electrophoretic protocols. Protein gels were subsequently stained with InstantBlue Coomassie stain (Expedeon) or transferred onto 0.2 μm PVDF membranes (Amersham Hybond P) using a Bio-Rad Trans-blot Turbo Transfer system. After a blocking step with 5% skim milk in TPBS (0.05% Tween-20 in 1× PBS), most immunoblots were probed with mouse primary antibodies (anti-FLAG M2 antibody (dilution 1:1000, Sigma-Aldrich #F1804), anti-STREP tag II antibody (dilution 1:1000, IBA Life Sciences #2–1508–050), and anti-polyhistidine antibody (dilution 1:2000, Sigma-Aldrich #H1029)) and a secondary anti-mouse antibody coupled to horseradish peroxidase (dilution 1:10000, Sigma-Aldrich #A4416) and visualized on film using Amersham ECL detection reagents. HA tag detection was performed with an HRP-conjugated anti-HA antibody (dilution 1:1000, Roche #12013819001) and visualized similarly. All western blots for subunit identification were performed on density gradient-purified complexes to assure proper complex assembly upon epitope tag introduction. For the dot-blot assays in Supplementary Fig. 2, transformed *E. coli* BL21 (DE3) cells were grown at 37 °C in antibiotics-supplemented liquid TB medium to $OD_{600}$ of 1, transferred to 17 °C, and induced with 0.7 mM IPTG for 16 h of protein expression. 1094 strains were grown in M63BI medium at 37 °C to $OD_{600} = 0.6$. Cells were pelleted by centrifugation (4000×g, 20 min, 4 °C), the supernatant was carefully blotted off and the wet weight of cells was measured. Cells were subsequently resuspended in buffer (2 ml of 120 mM NaCl, 20 mM HEPES pH 8 added per gram wet cells) and solubilized by the addition of SDS to a final concentration of 10%. Viscosity was reduced by brief sonication. Overall, 4 μl of each sample was spotted on 0.2 μm nitrocellulose membrane (Amersham Protran), allowed to dry and probed by standard immunostaining protocol. For the dot-blot assays in Supplementary Fig. 7, *E. coli* BL21 (DE3) cells were similarly grown, induced, and collected as above, and were subsequently solubilized in buffer containing 5% SDS, 120 mM NaCl and 20 mM HEPES pH 8. The resultant lysates were further diluted with detergent-free buffer to final SDS concentration of 2%. Overall, 2 μl of lysate, corresponding to 20 μl of induced culture at $OD_{600} = 1$, were spotted on 0.2 μm nitrocellulose membrane (Amersham Protran), allowed to dry and probed by immunostaining.

**In vitro assay of de novo cellulose biosynthesis**. Following parallel anti-FLAG affinity purification of the tested variants as above, elution fractions were concentrated on 100 kDa cut-off Amicon® Ultra centrifugal filters (Merck Millipore) and normalized for total protein content based on absorbance at 280 nm. A total of 100 μl of 0.1 mg ml⁻¹ samples were then incubated with 10 mM EDTA for 15 min on ice and buffer-exchanged on PD-10 columns in desalting buffer (20 mM HEPES pH 8, 120 mM NaCl, 10% glycerol, 0.008% LM-NPG, 2 μM c-di-GMP, and 1 tablet per 50 ml complete EDTA-free protease inhibitors) to a final volume of 1 ml. To each sample were added ATP and $MgCl_2$ to final concentrations of 5 mM and samples were split in two. For de novo cellulose synthesis, 5 mM UDP-glucose (Sigma-Aldrich) were added to the first half of each sample, whereas the other was supplemented with an equal volume of desalting buffer. Following reaction incubation at 30 °C for 45 min, equal volumes of each sample were transferred to 96-well polysorbent plates (MaxiSorp, Nunc) and incubated for 2 h at 4 °C. Then, wells were blocked with 2% skim milk in TPBS, followed by incubation with rabbit anti-cellulose primary antibody (dilution 1:100, custom-raised in rabbit using cello-hexaose (Sigma-Aldrich) conjugated to BSA) and anti-rabbit HRP-conjugated secondary antibody (1:3000, Cell Signaling Technology #7074). Substrate reagent pack (R&D Systems) was used for signal development and detection was carried out in an Infinite M200 PRO plate reader (TECAN) at wavelength of 420 nm, with subtraction wavelength of 525 nm. Wild-type and Δbcs *E. coli* 1094 cultures were used as positive and negative controls, respectively.

**Bacterial two-hybrid complementation assay**. To construct the recombinant plasmids used in the bacterial two-hybrid complementation assays, genes coding for the different Bcs proteins were amplified by PCR using genomic DNA from wild-type or $bcsQ^{G14S}$ *E. coli* 1094 strains as template. Amplified DNA fragments were digested with BamHI/KpnI (for bcsQ, bcsR, bcsF, and bcsG) or with BamHI/SacI (for bcsE), and subcloned into the similarly digested pKT25 and pUT18C vectors[19]. The resulting recombinant plasmids expressed hybrid proteins, with the proteins of interest represented as C-terminal fusions to the T25 or T18 fragments of the bacterial adenylate cyclase[19]. Cloning of BcsA subfragments and BcsE GIL domain was performed using the Gibson assembly procedure[40]. Gibson fragments were cloned into the pUT18 or pUT18C vectors as either N-terminal (for the N-terminal domain and the gating loop fragment) or C-terminal (for the glycosyl transferase, PilZ domain of BcsA and GIL domain of BcsE) fusions to the T18 fragment of the adenylate cyclase, respectively. These plasmids were used as templates to introduce point mutations in the conserved $R^{696}xxxR$ motif within the PilZ domain of BcsA (changed into AxxxR, RxxxA, AxxxA) and the $R^{415}xxD$ motif within the GIL domain of BcsE (changed into DxxD, DxxA, AxxA, and AxxD in the $BcsE^{GIL}$ construct and into DxxA in the full-length BcsE fusion). All constructs were verified by DNA sequencing.

The adenylate cyclase complementation assay was performed as previously published[19]. Briefly, the *E. coli* cya BTH101 strain was co-tranformed with derivatives of the pUT18c/pUT18 and pKT25 plasmids and co-transformants were plated on LB-agar containing 40 μg ml⁻¹ X-gal, 0.5 mM IPTG and appropriate antibiotics. Protein interactions were evaluated by blue colony color. In separate, more stringent experiments, colonies from each co-transformation were further resuspended in minimal media and spotted on M63B1-agar plates supplemented with 0.4% maltose, 40 μg ml⁻¹ X-gal, 0.5 mM IPTG and antibiotics. Protein interactions were evaluated by both growth and blue colony color.

**Data availability**. The electron microscopy reconstructions have been deposited in the Electron Microscopy Data Bank (EMDB) under accession codes EMD-3864 for the Bcs macrocomplex and EMD-3877 for the inner membrane subcomplex, respectively. The data that support the findings of this study are available from the corresponding authors upon request.

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

## Acknowledgements

We thank Eric Durand, Nicolas Reyes, Holger Sondermann, and Vincent Lee for useful discussions, expression constructs, and/or critical reading of the manuscript, and Annick Dujeancourt for indirect technical assistance. This work was supported by the *Institut Pasteur*, the Institute for Integrative Biology of the Cell (I2BC), the Centre National de la Recherche Scientifique (CNRS), the French Government's Investissement d'Avenir program: Laboratoire d'Excellence "Integrative Biology of Emerging Infectious Diseases" (Grant No. ANR-10-LABX-62-IBEID to J.-M.G.), the Fondation pour la Recherche Médicale (Grant No. DEQ20140329508 to J.-M.G.), the European Research Council (Grant MolStructTransfo to R.F.), and the ATIP-Avenir 2016 Program (to P.V.K.). In addition, P.V.K. was the recipient of a post-doctoral "Roux-Cantarini" fellowship from the Institut Pasteur and J.B.-B. was the recipient of a long-term post-doctoral fellowship from the Federation of European Biochemical Societies (FEBS).

## Author contributions

J.-M.G., P.V.K., J.B.-B. and R.F. conceived the project and P.V.K., J.B.-B., J.-M.G. and R.F. designed the experiments. All strains and expression constructs new to the study were generated by J.B.-B., L.T. and F.A.M. in the 1094 genetic background and by P.V.K. for heterologous expression in BL21 (DE3) cells. L.T. and F.A.M. performed the initial pulldowns for the proteomic identification of Bcs-interacting partners. P.V.K. optimized the expression and purification protocols for structural studies, analyzed the purified complexes, collected and, together with R.F., analyzed EM data. B2H experiments were designed by J.B.-B. and P.V.K. and performed by J.B.-B. J.B.-B. performed the in vivo activity assays of calcofluor-binding and P.V.K. and J.B.-B. designed, and performed the in vitro activity assays of de novo cellulose synthesis. P.-A.K. and G.K. contributed to this work with discussions, constructs generation, and/or preliminary experiments. P.V.K. and J.-M.G. wrote the paper with significant contributions from J.B.-B. and R.F. and feedback from all authors.

## Additional information

**Competing interests:** The authors declare no competing financial interests.

