## [Peer Review File · Nature Communications]

Reviewers' comments:

Reviewer #1 (Remarks to the Author):

The authors present a work aimed at gaining insights into the structure and assembly of a sub-complex of the E.coli cellulose secretion system. They use an E.coli cellulose-secreting strain as a model and show that Bcs proteins of inner membrane and the cytosol interact. Using chemical cross-linking they stabilise a macromolecular complex that is formed by an undefined number of subunits of BcsA, E,F,Q,R and six subunits of BcsB.

By heterologous reconstitution and mutational analyses they propose a structure-function model, in which they suggest accessory cytosolic proteins to regulate cellulose biogenesis by affecting both the assembly and stability of the inner membrane BscA/B complex.

In summary, the current work and especially it's interpretation would significantly benefit by:

(1) demonstrating rigorous validation for EM reconstructions (is the handedness correct?)

—random-conical-tilt analysis

—and tilt-pair validations

— reprojections etc.

At this point in time, the correctness of the structure is to be shown.

(2) analysing the topological arrangement of all proteins involved (which part of the structure is resembled by which protein?)

— labelings using fusion proteins and/or specific gold labels

(3) providing data about the protein stoichiometry of the complex/es

— tandem-affinity purifications

— quantitative mass spec

Specific comments:

- Why didn't the authors attempt to obtain structures by cryoEM? The negatively stained particles appear very promising!

- As described at page 23, line 808, volumetric analysis on the EM map with the software UCSF Chimera was used to assess the MW of the complex. Considering that this measurement (0.81 Da/A³) is performed on a EM map obtained from negatively stained particles and especially at low resolution, it is highly likely that the measurement is incorrect.

- The authors state the protein-complex is formed by multiple copies of the individual components, but only for the BcsB they are able to give the exact number of subunits by a tentatively fitting a crystal structure-based model in the EM map. Why was BcsA not fitted in the EM map if both maps and x-ray structures are available?

- Page 5 line 156, and Figure 3 panel D, it is not clear how the authors justify the presence of the additional density mentioned as “the stump”. The way this is explained is too vague and not clear in the text. Gold labeling could be helpful to identify the component forming this density.

- Provide a summary table for all the structures: Since six independent single particle analyses were performed, a table including the maps/classes of the six computations, together with the genes and protein components as shown in Figure 1A and 1D could be useful to summarise the different samples.

- Figure 2B, consider adding a scale bar.

Reviewer #2 (Remarks to the Author):

Krasteva et al “Insights into the structure and assembly of a bacterial cellulose secretion system”

In this paper the authors successfully map out interactions between the various proteins in the bacterial cellulose secretion system and present a negative stain EM reconstruction of the inner membrane machinery involved in cellulose biosynthesis at 17 Å resolution. Despite the limited resolution available as a consequence of using negative stain EM rather than cryo-EM, this is an elegant piece of work that significantly advances the field.

Specific Comments:

1) Line 47: “scaffolding” typo (should read scaffolding).

2) Lines 124-126: bcsZC was not seen in pull-downs with the macrocomplex expression in BL21(DE3) cells. Was this before or after purification with the cross-linking agent via gradient centrifugation? Perhaps these periplasmic/outer membrane components form transient complexes, and some mild cross-linking could provide extra density via EM? What is the

significance to the absence of these proteins?

3) Lines 138-142: It would be useful if the authors could provide quantitative data on the relative abundance of the components by, for example, using densitometry. Western blots using the same tag should be provided on the same blot, not separate strips where it's hard to determine if the background contrast is the same.

4) Lines 160-164: It is noted by the authors that BcsEFG are likely involved in regulation of the stability of the complex and less so in oligomerization. Presumably these are loosely associated with the subcomplex? If they are not coexpressed with BcsAB, a stable complex is still formed and the structure was determined. Thus, can the authors comment on whether BcsEFG really necessary for stabilization of the multimeric complex?

5) Lines 222 - 226 - It's a bit of a stretch to suggest that just because an interaction between the PilZ domain of BcsA and BcsE was observed by Bac2H that c-di-GMP can bind at the interaction interface between these two proteins and that this is involved in stabilizing the secretion complex. In the case of PgaCD, the c-di-GMP binding domain is formed at the interface of the two proteins and both contributed to binding in a co-operative way, while BcsA and BcsE both have separate c-di-GMP binding folds. Also, we know from structural studies of BcsAB that binding of c-di-GMP to the PilZ domain relieves an auto-inhibited state of BcsA, providing a rationale for why c-di-GMP is required for cellulose biosynthesis by this complex. The authors should explore mutations in the PilZ domain of BcsA or the RxxD motif of BcsE that prevent c-di-GMP binding to observe the effect on the stability/formation of the Bcs macrocomplex to substantiate this statement. Perhaps the presence of BcsE, rather than its binding of c-di-GMP specifically, is all that's required for the observed stability.

6) Lines 226 - 229 - Is there precedence for GIL-like proteins (or any c-di-GMP binding fold for that matter) to serve as a reservoir of local c-di-GMP? Can the authors please comment on how the proposed nucleotide relay between BcsE and BcsA would occur? Is it known whether there are phosphodiesterases localized to the Bcs complex that would necessitate protection of c-di-GMP by BcsE?

7) Lines 232-233: The interactions between the N-terminal domain of BcsA and BcsG are described as "strong". Please can the authors provide the data to substantiate this claim of "strength". Presumably, a strong interaction would have been resulted in BcsG forming part of the inner membrane subcomplex. Perhaps this interaction is more transient than otherwise thought?

8) The authors should perhaps emphasize and compare further the importance of this structure of the in vivo complex as seen in the 1904 strain supported with non-cellulose secreting strains in

comparison to the in vitro BcsA/B structure from reference 17. Were there any indications from the BcsAB crystal structure that would suggest the multimeric subcomplex seen here via cryo-EM? Can the crystal structure be modeled further into the EM envelope?

9) Lines 248 - 257 - A reference to Figure 4b needs to be provided somewhere in here.

10) Lines 253-257: The open, non-cylindrical architecture observed is proposed by the authors to allow for post-synthetic modifications by BcsZ. Is it possible for BcsZ to function “in between” synthesis and export, rather than “alongside”? Is it known if BcsZ interactions with multimeric vs monomeric BcsAB? Is this open, non-cylindrical architecture observed in other systems with or without modifications to the polymer?

11) State the consequence of the G14S mutation in BcsQ in the text.

Figures:

12) Figure 1b - BcsQ and BcsB are coloured the same, but to my knowledge BcsB is not required for polar localization and is not a MinD homolog. Accessory subunits BcsFR and BcsG are coloured differently, but are not represented in the figure key.

13) Figure 1c - A negative control for the calcofluor binding panel would help in interpretation of the data.

14) Figure 3A: It would be nice if this model was further expanded to include the modelled structure of BcsA for clarity and to illustrate the authors' note on lines 156-157 of additional density that can not be entirely attributed to BcsA. This would help to clarify Extended Figure 5i, and to potentially map where some of the interactions may be taking place (adding to the direct pull-down interaction data). A final model of individual protein structures in the overall EM density, even if just a hypothetical model, could also help to illustrate the importance of BcsB C-terminal domains for assembly of complex formation and BcsA synthase activity. Is it possible to delete domains of BcsA or BcsB to elucidate positioning and further map out or model the individual proteins in the envelope?

15) Figure 4b - In the left hand diagram, how would BcsC assemble to form a composite outer membrane beta-barrel, which would presumably be symmetric, while the TPR domains remain asymmetric to align with the Bcs macrocomplex and allow access to the polymer for BcsZ? Generally, if each Bcs macrocomplex is generating six separate cellulose polymers, would there not need to be six BcsA subunits to catalyze this? Does the approximate size and molecular weight of the macrocomplex support the presence of six BcsA subunits?

16) Ext. Data Figure 1b - Can you complement the mutations to show that the loss of calcofluor binding is due to the deletion of the gene specifically and not polar effects?

Reviewer #3 (Remarks to the Author):

This is a very interesting study that describes the overall composition and low-resolution architecture (from single-particle EM) of the multiprotein complex required for cellulose biosynthesis in *E. coli*. Key elements of the complex are conserved across a wide range of bacteria, where cellulose participates in biofilm formation. In addition, other bacterial EPSs are assembled using similar principles, so the observations and experimental approaches have broader relevance. The work is comprehensive and clearly presented. It now lays the foundation for higher-resolution structures and identifies questions that can be tested by biochemical approaches.

Scientific Comments:

1. How do you explain the distribution of BcsQ into both soluble and membrane fractions when the membrane-bound form cannot be extracted without harsh treatment?
2. In the pull-down experiments, it is curious that cytoplasmic BcsQ pulls down OM BcsC but IM BcsA does not. Presumably, the amount of BcsC is small since it does not show in the SDS-PAGE profiles. Please comment.
3. Three irrelevant proteins are purified in pull downs in amounts sufficient for detection on stained gels. Please comment on the normal cellular location of McsK, AtpA and AceE. Also, with the higher sensitivity of LC-MS, are any other contaminating proteins detected?
4. In the later experiments, a modified operon is used with the goal of overexpression. Was the complex really overexpressed in these conditions, relative to the WT format? If so, to what extent?
5. Are cellulose synthase activity values for the various subcomplexes?
6. Is it possible to generate any sense of the stoichiometry of the various proteins from the stained gel profiles and do the homotypic interaction results help?
7. Can you please explain the origin of the eclectic detergent mixture used to extract the complex – is it from a screen or past precedent (in which case, please add a reference)?

Editorial Comments:

Line 55: I don't think "unique" is needed here; it might be an architecture not seen before but we don't yet know if will be unique.

Line 72: add a ref for the statement about BcsE.

Fig 1C: I think the calcofluor colonies need a negative control for non-experts. I know these controls appear in Extended Data Fig 1 but some readers may not want to dig that deep.

Line 81: please define "crude" solubilization. I assume this is the same protocol described in extended methods?

Line 100: please define "mild" chemical crosslinking.

Line 108: I do not think "remarkably" is needed here. That this is a new structure shouldn't be surprising and the significance of the finding is clear.

Line 120: does BL21 lack the bcs genes, or is it unable to make cellulose because of a mutation?

Line 145: another example where "remarkably" seems unnecessary.

Extended Data Fig. 7. Please explain in the legend the relevance of the Zip controls for non-experts.

Extended Data Fig. 8b. This can be updated with the T2SS translocon structure that just appeared in NSMB.

Reviewers' comments:

Reviewer #1 (Remarks to the Author):

The authors present a work aimed at gaining insights into the structure and assembly of a sub-complex of the E.coli cellulose secretion system. They use an E.coli cellulose-secreting strain as a model and show that Bcs proteins of inner membrane and the cytosol interact. Using chemical cross-linking they stabilise a macromolecular complex that is formed by an undefined number of subunits of BcsA,E,F,Q,R and six subunits of BcsB.

By heterologous reconstitution and mutational analyses they propose a structure-function model, in which they suggest accessory cytosolic proteins to regulate cellulose biogenesis by affecting both the assembly and stability of the inner membrane BscA/B complex.

In summary, the current work and especially its interpretation would significantly benefit by:

- (1) demonstrating rigorous validation for EM reconstructions (is the handedness correct?)*
 - random-conical-tilt analysis*
 - and tilt-pair validations*
 - reprojections etc.*

At this point in time, the correctness of the structure is to be shown.

We collected a dataset collected in thicker negative stain in tilt pairs and obtained initial 3D models by random conical tilt reconstruction in EMAN 2.0 (Extended Data Fig. 4d). However, the low resolution of the raw negative-stain data, the application of a low-pass filter to reduce noise artifacts (cut at 25Å), the resulting very low resolution of the starting model (further cut at 60 Å prior to 3D classification and refinement) and the still not high resolution of the final refined model could not fully resolve the ambiguity in the RELION-assigned handedness. This, together with the fact that tilt data collection in negative stain is bound to be plagued by non-sample-specific density artifacts in the tilted mode due to stacking of the contrasting agent, led us to include a caveat in the results section allowing for flipped handedness in the final reconstruction.

We would like to underscore, however, that all initial models refined to a 3D model with the same distinctive features, and that high-contrast class-averages closely matched 2D reprojections by visual inspection (Extended Data Fig. 4a). In addition, all of the key structural and functional conclusions of our work (multimeric assembly, asymmetric architecture, strong association with accessory subunits, relative orientation across the membrane, assignment of electron density segments, etc.) can be drawn even at the level of 2D class averages and are not affected by the absolute handedness of the final model.

- (2) analysing the topological arrangement of all proteins involved (which part of the structure is resembled by which protein?)*
 - labelings using fusion proteins and/or specific gold labels*

We have determined the topological arrangement of the proteins to a degree that is on par with the structures' resolutions. Our study identifies for the first time multimeric assembly of an exopolysaccharide biosynthetic complex, and by structurally analyzing several multi-subunit complexes pinpoints the relative orientation of the macroassembly relative to the bacterial membrane, as well as the regions of the map corresponding to the periplasmic, inner-membrane and cytosolic components.

We cannot present exact localization of the relatively small cytosolic components as their individual size would be within the resolution error. Using bulky gold labels that would be visible in negative stain (i.e. 5nm gold particles) and adding fusion proteins and tags through extended flexible linkers to

the small cytosolic components would provide a localization estimate within more or less the same error as the multiple topology analyses already presented.

(3) providing data about the protein stoichiometry of the complex/es
— *tandem-affinity purifications*
— *quantitative mass spec*

We discuss the challenges of these studies below and why we have refrained from presenting such data.

Specific comments:

- Why didn't the authors attempt to obtain structures by cryoEM? The negatively stained particles appear very promising!

We agree with the reviewer that the single particles appear very promising and we are actively pursuing higher-resolution studies by cryo-EM. Several factors have so far interfered such as the relatively low yields of purified complexes, the high concentrations of glycerol used during the purification steps, the optimization of crosslinking protocols for work at high protein concentrations and absence of viscosity agents, etc.

We would like however to stress that very few of the secretion systems studied to date and highlighted in the extended data (Extended Data Fig. 9) can boast of high-resolution studies and that the method of choice has been primarily NS-EM.

For secretion systems, or any other multi-subunit macrocomplexes for that matter, high-resolution cryo-EM data have typically followed years of low-resolution studies and efforts from multiple laboratories. It has to be considered that our work is the first to describe the assembly and structure-functional role of the multiple accessory subunits acting in bacterial cellulose secretion and therefore we believe it will be of high interest even at the lower resolution that negatively stained data can provide.

- As described at page 23, line 808, volumetric analysis on the EM map with the software UCSF Chimera was used to assess the MW of the complex. Considering that this measurement (0.81 Da/A3) is performed on a EM map obtained from negatively stained particles and especially at low resolution, it is highly likely that the measurement is incorrect.

We have provided an estimate for the molecular weight using volumetric measurements of the map volume and a widely accepted standard value for protein density (Harpaz *et al.*, 1994). Indeed, at low resolution and using negatively stained particles this estimate could be either over- or under-estimated due to stain not penetrating in solvent accessible cavities in the structure, protein density corresponding to flexible surface regions being averaged out during the reconstruction, contour level in Chimera being chosen arbitrarily, etc. However, such estimates are commonly accepted in the field and provide a rough estimate of the complexes' molecular weights that are on par with the rest of the structural conclusions.

We would like also to underscore that alternative approaches for molecular weight estimation also face severe challenges when dealing with such huge structures and where no high-resolution data is available. For example, methods such as size-exclusion chromatography and velocity analytical ultracentrifugation are dependent on hydrodynamic radii and can be skewed with non-globular samples, whereas more quantitative techniques such as MALS, native MS, and equilibrium analytical centrifugation are extremely difficult to optimize due to, again, the large complex size, and also the presence of a mix of detergents, viscosity agents, high-absorbance nucleotides and traces of contaminating proteins of unknown quaternary structure that can interfere with signal detection.

We therefore believe that giving a rough estimate based on single-particle reconstruction of the assembled Bcs complex could be of value, albeit not absolutely precise. We have however clarified and corrected key aspects of the molecular weight estimates: i) During the revision, we realized that the voxel size of the initial models has been likely overestimated by less than 5% during microscope calibration and have corrected all models to what is in agreement with published structures; ii) We have underscored both in the materials and methods section and in the figure legends that the presented molecular weights are only estimates based on the low-resolution envelope reconstruction and not the intrinsic to the proteins density (line 400 and line 931); iii) We have adjusted the volume threshold based on the density assigned to the periplasmic modules of hexameric BcsB and used the standard value for volume to mass conversion; iv) We have referred to the complex as simply “megadalton-sized” in the abstract and body of the main text (line 40, line 116 and line 400).

- The authors state the protein-complex is formed by multiple copies of the individual components, but only for the BcsB they are able to give the exact number of subunits by a tentatively fitting a crystal structure-based model in the EM map. Why was BcsA not fitted in the EM map if both maps and x-ray structures are available?

We have previously refrained from fitting BcsA subunits due to the multidomain organization of the protein as well as the presence of an additional N-terminal module relative to the known structure. While the localization of the transmembrane domain can be inferred from the relative position of the BcsB periplasmic module, both the glycosyl transferase domain and the PilZ domains feature long linkers proximal to their N-terminal borders that can allow for significant conformational restructuring in the context of the multisubunit complex and relative to the crystal structures of the *Rhodobacter* BcsAB duo. Given the low resolution of the reconstruction we prefer therefore to avoid over-interpretation of our data by extrapolating the homologous crystal structure to our system and therefore now present only partial fitting of the BcsA transmembrane regions in the M-tier density (Extended Data Fig. 5j).

- Page 5 line 156, and Figure 3 panel D, it is not clear how the authors justify the presence of the additional density mentioned as “the stump”. The way this is explained is too vague and not clear in the text. Gold labeling could be helpful to identify the component forming this density.

In the absence of the second *bcsEFG* operon, we observe both limited proteolysis within BcsA and substoichiometric quantities of BcsQ and BcsR that co-purify with the complex. This would suggest that although the membrane subunits assemble in their characteristic multimeric architecture, we likely have compromised stability and heterogeneity at the level of the periplasmic modules. As the ‘stump’ is observed at the level of 2D class averages and consequently the 3D reconstruction we are inclined to concur that it likely does not correspond to a specific module but rather to residual density of the cytosolic BcsA and possibly BcsR/Q domains after averaging-out conformational heterogeneity.

We have modified the text to clarify this discussion (lines 180-184).

- Provide a summary table for all the structures: Since six independent single particle analyses were performed, a table including the maps/classes of the six computations, together with the genes and protein components as shown in Figure 1A and 1D could be useful to summarize the different samples.

We have now included a table summarizing the different datasets (Extended Data Fig. 6k).

- Figure 2B, consider adding a scale bar.

We have added a scale bar to all micrographs in the main text and extended data figures (Figure 2b and Extended Data Fig. 4b).

Reviewer #2 (Remarks to the Author):

Krasteva et al "Insights into the structure and assembly of a bacterial cellulose secretion system"

In this paper the authors successfully map out interactions between the various proteins in the bacterial cellulose secretion system and present a negative strain EM reconstruction of the inner membrane machinery involved in cellulose biosynthesis at 17 Å resolution. Despite the limited resolution available as a consequence of using negative stain EM rather than cryo-EM, this is an elegant piece of work that significantly advances the field.

Specific Comments:

1) Line 47: "scaffolding" typo (should read scaffolding).

The text has been corrected and reads "scaffolding" in the revised submission (line 61).

2) Lines 124-126: bcsZC was not seen in pull-downs with the macrocomplex expression in BL21(DE3) cells. Was this before or after purification with the cross-linking agent via gradient centrifugation? Perhaps these periplasmic/outer membrane components form transient complexes, and some mild cross-linking could provide extra density via EM? What is the significance to the absence of these proteins?

Indeed, the BcsZ and BcsC components did not co-purify with the Bcs complex used for structural studies and were not visible on the SDS-PAGE gel following elution from the anti-FLAG resin. It is important to note that the complex used for structural studies was extracted with high-concentrations of a mix of low-CMC detergents. The crosslinking agent was used in a very low concentration and only during gradient purification in a single context (low-yield, constitutive, chromosome-driven expression in the 1094 genetic background), and after stringent washes of the affinity resin (50-100 column volumes) and elution of the stably associated fraction. The experimental procedures for each purification have been detailed in the methods section.

Whereas we are certainly interested in mild *in vivo* crosslinking and stabilization of the fully assembled secretion system, what is presented in the manuscript are results from thorough optimization of the protocol toward sample homogeneity and at this stage we simply cannot present data on a fully assembled secretion system encompassing all Bcs components. In addition, we would like to note that the addition of both a non-hydrolyzable ATP homologue and c-di-GMP to the purified system visibly improved sample quality in terms of homogeneity and is likely to stabilize a particular state of the system's reaction cycle, which might not be necessarily a high-affinity interacting partner for the BcsZ and/or BcsC components in the absence of processively secreted cellulose.

Finally, when BcsA^{HA-FLAG} was used as bait in the context of relatively milder extraction (0.1% Triton X-100) for initial detection of Bcs interacting partners by MS, BcsZ and BcsC were not detected either. BcsZ was not detected even in the context of affinity purification through BcsQ and this is in line with both its structure of fully soluble, single-domain periplasmic protein and our model of open system's architecture.

In contrast, a few BcsC peptides were detected when overexpressed and tagged BcsQ was used as bait. The sample, however, featured very high heterogeneity for structural studies likely due to non-stoichiometric BcsQ expression in the specific genetic background and/or formation of multiple species of partially assembled complexes.

We therefore present data only on the inner membrane and cytosolic components although hoping that we will be able to capture a homogeneous and fully assembled system at a later point.

3) Lines 138-142: It would be useful if the authors could provide quantitative data on the relative abundance of the components by, for example, using densitometry. Western blots using the same tag should be provided on the same blot, not separate strips where it's hard to determine if the background contrast is the same.

We have refrained from doing densitometry analysis due to the co-migration of various complex components and as well as with trace amounts of co-purified but likely irrelevant proteins. Western blot analyses were performed on multiple different complexes the expression levels and assembly stability of which vary depending on the subunit tagged. In addition, western blot transfer efficiencies are also protein-dependent with the low-molecular weight BcsR and the SDS-resistant lower band of BcsA being somewhat challenging to transfer. Although it is certainly possible to adjust amounts loaded on the gel for the sake of presentation, the relatively low yields, long preparation protocols and exorbitant costs of the detergents, nucleotides, etc. used for extraction and purification of the various complexes do not justify further optimizing the data representation.

4) Lines 160-164: It is noted by the authors that bcsEFG are likely involved in regulation of the stability of the complex and less so in oligomerization. Presumably these are loosely associated with the subcomplex? If they are not coexpressed with BcsAB, a stable complex is still formed and the structure was determined. Thus, can the authors comment on whether BcsEFG really necessary for stabilization of the multimeric complex?

Indeed, the BcsRQAB subunits are sufficient for expression of the inner membrane subunits, as well as their oligomerization and assembly of the upper membrane platform and periplasmic crown repeats. However, the complex purified in this context features both limited BcsA proteolysis and substoichiometric BcsRQ association relative to the Bcs macrocomplex purified upon co-expression of BcsEFG (Figure 3d). The lack of the BcsEFG or BcsFG subunits leads to only residual density in the region that would correspond to the cytosolic modules (i.e. the 'stump'), likely due to dissociation, proteolysis and conformational heterogeneity that is affecting the cytosolic modules and is averaged out during image processing. Therefore, we believe that we are correct in the conclusion that the *bcsEFG* operon does not determine the oligomerization of the system *per se* (i.e. the BcsA/B duo gets into the membrane and oligomerizes) but contributes to cellulose biogenesis by regulating the stability of the multimeric synthase complex (BcsA is protected from limited proteolysis and BcsRQ are kept in place).

5) Lines 222 - 226 - It's a bit of a stretch to suggest that just because an interaction between the PilZ domain of BcsA and BcsE was observed by Bac2H that c-di-GMP can bind at the interaction interface between these two proteins and that this is involved in stabilizing the secretion complex. In the case of PgaCD, the c-di-GMP binding domain is formed at the interface of the two proteins and both contributed to binding in a co-operative way, while BcsA and BcsE both have separate c-di-GMP binding folds. Also, we know from structural studies of BcsAB that binding of c-di-GMP to the PilZ domain relieves an auto-inhibited state of BcsA, providing a rationale for why c-di-GMP is required for cellulose biosynthesis by this complex. The authors should explore mutations in the PilZ domain of BcsA or the RxxD motif of BcsE that prevent c-di-GMP binding to observe the effect on the stability/formation of the Bcs macrocomplex to substantiate this statement. Perhaps the presence of BcsE, rather than its binding of c-di-GMP specifically, is all that's required for the observed stability.

We have explored such mutations which are now presented in the extended data. In line with our discussion, we observe I-site dependent interactions of both BcsE and the PilZ module of BcsA with subunits the complex (lines 244-266; Extended Data Fig. 8e-f). We know that the studied system of *Rhodobacter* lacks a second c-di-GMP sensor and that no multimeric assembly for that system has been reported to date. It is therefore highly likely that the *E.coli/Salmonella* system features a more sophisticated regulation of assembly and function, which occurs at least partly at the level of c-di-GMP sensing through multiple modules. We have discussed how a possible signal relay can occur structurally in our next rebuttal comment. However, the full system activation mechanism and the exact role of c-di-GMP sensing and enzymatic activities of the accessory Bcs subunits is yet to be explored.

6) Lines 226 - 229 - Is there precedence for GIL-like proteins (or any c-di-GMP binding fold for that matter) to serve as a reservoir of local c-di-GMP? Can the authors please comment on how the proposed nucleotide relay between BcsE and BcsA would occur? Is it known whether there are phosphodiesterases localized to the Bcs complex that would necessitate protection of c-di-GMP by BcsE?

To our knowledge, BcsE is the only GIL domain protein that has been studied to date. However, its c-di-GMP binding site is proposed to be similar to the c-di-GMP binding I-site of diguanylate cyclases (GIL stands for GGDEF I-site-like), which in other systems has been shown to act in a nucleotide signal relay to affect downstream signaling proteins. A recent example is the Lap system in *P. fluorescens*, where the diguanylate cyclase GcbC directly interacts with LapD in an I-site dependent manner to regulate surface adhesin maintenance through the LapD/G/A system (Dahlstrom et al., 2015, Dahlstrom et al. 2016).

Here we see similarly an I-site-dependent interaction of BcsE with BcsQ, a central of the secretion system, which in turn also interacts with the c-di-GMP sensing PilZ module of the catalytic subunit (Extended Data Fig. 8e-f). The latter interaction does not depend on the integrity of the c-di-GMP binding PilZ domain linker, which indicates that the binding site is likely exposed in the context of the binary BcsA^{PilZ}-BcsQ interaction. Even more importantly, BcsE also interacts directly with the PilZ module of BcsA and this interaction depends on the integrity of both c-di-GMP binding sites (Extended Data Figure 8e-f).

C-di-GMP binds in the form of an intercalated dimer in all structures of c-di-GMP-bound diguanylate cyclases and in the c-di-GMP-bound structures of the *Rhodobacter* BcsA homolog, leaving a copy c-di-GMP exposed and largely free from protein-coordinating contacts. It is therefore plausible, that dimeric c-di-GMP bound at the interface of BcsQ/BcsE can expose a copy of c-di-GMP to be sensed by the PilZ module of BcsA, which is held in immediate proximity by BcsQ.

Regarding the potential role of a phosphodiesterase, we are not aware at this point of the presence of such phosphodiesterase. However, *E. coli/Salmonella* cellulose secretion is regulated by specific diguanylate cyclases (AdrA and/or YedQ), which can explain the co-evolution of a dedicated set of c-di-GMP receptors.

We have included part of this discussion in the text as well as the caveat that it requires further examination (lines 244-266).

7) Lines 232-233: The interactions between the N-terminal domain of BcsA and BcsG are described as “strong”. Please can the authors provide the data to substantiate this claim of “strength”. Presumably, a strong interaction would have been resulted in BcsG forming part of the inner membrane subcomplex. Perhaps this interaction is more transient than otherwise thought?

We thank the reviewer for this comment and have corrected the text to avoid over-interpretation of indirect data. While the signal of adenylate cyclase functional reconstitution, as judged by X-gal

hydrolysis and the resulting blue color, is more pronounced for this particular B2H pair, we have indeed no indication of the strength of interaction between the two modules in their native functional context. In fact, differences of B2H intensities can be a product of multiple factors apart from strength of interaction, such as fusion protein stability, preferred oligomerization state, steric hindrances within the complex, etc. As the presence but not the specific strength of interaction is what matters from the experiment and in terms of discussion, we have omitted 'strong' from the corresponding sentence (line 256).

8) The authors should perhaps emphasize and compare further the importance of this structure of the in vivo complex as seen in the 1904 strain supported with non-cellulose secreting strains in comparison to the in vitro BcsA/B structure from reference 17. Were there any indications from the BcsAB crystal structure that would suggest the multimeric subcomplex seen here via cryo-EM? Can the crystal structure be modeled further into the EM envelope?

We now present an extended data figure panel, which demonstrates the crystal packing of structures of the *R. sphaeroides* BcsA-BcsB complex in different deposited structures, which correspond to various reaction cycle states (Extended Data Fig. 5a-b). There are two preferential types of crystal packing: in the first one, adjacent BcsAB complexes pack in an antiparallel fashion; in the second, the BcsAB complexes arrange in a side-by-side fashion without significant contacts between the membrane modules. Taken together, these data rule out multimerization insights that could be gained from earlier studies.

We have briefly discussed this in the revised text (lines 132-134).

9) Lines 248 - 257 - A reference to Figure 4b needs to be provided somewhere in here.

We have referred to the two models represented in figure Figure 4b along with their discussion in the paragraph (lines 291 and 293).

10) Lines 253-257: The open, non-cylindrical architecture observed is proposed by the authors to allow for post-synthetic modifications by BcsZ. Is it possible for BcsZ to function "in between" synthesis and export, rather than "alongside"? Is it known if BcsZ interactions with multimeric vs monomeric BcsAB? Is this open, non-cylindrical architecture observed in other systems with or without modifications to the polymer?

We are not aware of other synthase-dependent systems for hydrolase-modified exopolysaccharide secretion to have been described structurally at the level of assembled multicomponent machinery, which is to underscore the novelty of this work. While there are some crystallographic studies of individual periplasmic components from the Pel, alginate and cellulose secretion pathways, there is no firm indication of whether the export occurs through a one-step or two-step secretion mechanism. However, given that no periplasmic accumulation of cellulose has been, to our knowledge, observed previously and that our earlier work has demonstrated both polar BcsQ localization and polar secretion of relatively thick cellulose fibers, we favor models for cellulose secretion that occurs processively through the double envelope with parallel function of BcsZ.

11) State the consequence of the G14S mutation in BcsQ in the text.

The consequences of this mutation are indicated in the text (line 211 and lines 236-241).

Figures:

12) Figure 1b - BcsQ and BcsB are coloured the same, but to my knowledge BcsB is not required for

polar localization and is not a MinD homolog. Accessory subunits BcsFR and BcsG are coloured differently, but are not represented in the figure key.

We have adjusted the colors to better distinguish between BcsQ (bright blue) and BcsB (purple). Detailed domain architectures for all subunits are represented in the extended data. This is now referred to in the Fig. 1 legend (lines 380-381).

13) Figure 1c - A negative control for the calcofluor binding panel would help in interpretation of the data.

We now present a negative control for the calcofluor binding panel by providing data for a Δbcs strain where the *bcs* cluster is deleted (Fig. 1c; Extended Data Fig. 1c).

14) Figure 3A: It would be nice if this model was further expanded to include the modelled structure of BcsA for clarity and to illustrate the authors' note on lines 156-157 of additional density that cannot be entirely attributed to BcsA. This would help to clarify Extended Figure 5i, and to potentially map where some of the interactions may be taking place (adding to the direct pull-down interaction data). A final model of individual protein structures in the overall EM density, even if just a hypothetical model, could also help to illustrate the importance of BcsB C-terminal domains for assembly of complex formation and BcsA synthase activity. Is it possible to delete domains of BcsA or BcsB to elucidate positioning and further map out or model the individual proteins in the envelope?

As discussed above, we have previously refrained from fitting BcsA subunits due to the multidomain organization of the protein as well as the presence of a small N-terminal module relative to the known structure. While the localization of the transmembrane domain can be inferred from the relative position of the BcsB periplasmic module, both the glycosyl transferase domain and the PilZ domains feature long linkers proximal to their N-terminal borders that can allow for conformational restructuring in the context of the multisubunit complex and relative to the crystal structures of the *Rhodobacter* BcsAB duo. Given the low resolution of the reconstruction we prefer therefore to avoid over-interpretation of our data by extrapolating the homologous crystal structure to our seemingly different system and therefore now present only partial fitting of the BcsA transmembrane regions in the M-tier density (Extended Data Fig. 5j). As seen in the corresponding extended data figure, the M-tier density can indeed fit 6 BcsA transmembrane regions.

15) Figure 4b - In the left-hand diagram, how would BcsC assemble to form a composite outer membrane beta-barrel, which would presumably be symmetric, while the TPR domains remain asymmetric to align with the Bcs macrocomplex and allow access to the polymer for BcsZ? Generally, if each Bcs macrocomplex is generating six separate cellulose polymers, would there not need to be six BcsA subunits to catalyze this? Does the approximate size and molecular weight of the macrocomplex support the presence of six BcsA subunits?

These are all very good questions. Regarding BcsC, there is a stretch of ~10 amino acids that are likely to form a flexible linker (up to ~3.5 nm in extended form) at what is predicted to be the membrane-proximal interdomain boundary (Extended Data Fig. 1a; lines 293-297). This can allow for significant conformational variability between the separate BcsC subunits and, potentially, for an asymmetric packing as in the proposed models. Regarding BcsA, the membrane module of the homologous structure folds in a flattened fold similar in lateral dimensions to the BcsB periplasmic module. So, at least speculatively, we can fit six BcsA transmembrane regions in the density, now shown in the extended data (Extended Data Fig. 5j).

16) Ext. Data Figure 1b - Can you complement the mutations to show that the loss of calcofluor binding is due to the deletion of the gene specifically and not polar effects?

We now show complementation data to demonstrate the lack of polar effects and have modified the text to underscore that the deletions are non-polar (Extended Data Fig. 1c; line 81).

Reviewer #3 (Remarks to the Author):

This is a very interesting study that describes the overall composition and low-resolution architecture (from single-particle EM) of the multiprotein complex required for cellulose biosynthesis in E. coli. Key elements of the complex are conserved across a wide range of bacteria, where cellulose participates in biofilm formation. In addition, other bacterial EPSs are assembled using similar principles, so the observations and experimental approaches have broader relevance. The work is comprehensive and clearly presented. It now lays the foundation for higher-resolution structures and identifies questions that can be tested by biochemical approaches.

Scientific Comments:

1. How do you explain the distribution of BcsQ into both soluble and membrane fractions when the membrane-bound form cannot be extracted without harsh treatment?

The complex purified for electron microscopy studies is membrane-extracted and demonstrates remarkably strong assembly affinity. Although we do not have quantitative data on this affinity, we can estimate that working concentrations for complex purification and observation are in the lower nanomolar range. This can explain the stable association with the membrane fraction requiring denaturing or detergent treatment to extract (Extended Data Fig. 2a). On the other hand, and as indicated in Figure 1, the expression strategy in the 1094 ^{FLAG-HA} *bcsQ* strain would lead to non-stoichiometric overexpression of some of the Bcs components and would explain excess cytosolic BcsQ, which is by prediction a soluble protein. We have changed the text to stress that BcsQ is overexpressed in this context (line 87).

2. In the pull-down experiments, it is curious that cytoplasmic BcsQ pulls down OM BcsC but IM BcsA does not. Presumably, the amount of BcsC is small since it does not show in the SDS-PAGE profiles. Please comment.

In fact, our data show that, although predicted as a soluble cytoplasmic protein, BcsQ is stably associated with the membrane within the Bcs complex. We agree that the pull-down of BcsC through Q is quite interesting but the sample featured extreme heterogeneity likely due to the non-stoichiometric overexpression genetic background, as discussed above and shown in Figure 1, and the formation of multiple species of partially assembled complexes. In contrast, BcsC was not detected in MS analyses of the total elution fraction when BcsA was used as bait (analysed in solution and not on cut-out bands), confirming that the affinity of the interaction is lower than what the working concentrations and resin washes would allow.

3. Three irrelevant proteins are purified in pull downs in amounts sufficient for detection on stained gels. Please comment on the normal cellular location of McsK, AtpA and AceE. Also, with the higher sensitivity of LC-MS, are any other contaminating proteins detected?

We are grateful to the reviewer for bringing our attention MS data as we noticed minor errors that we introduced during the integration of proteomics data from the different experiments (i.e. switching the names of two of the main contaminants) and omitting a fourth contaminant consistently detected in the preps (LpdA, a membrane, 52 kDa protein consistently identified in parallel with AtpA, 55kDa). We have now corrected these errors.

Of course, with the higher sensitivity of LC-MS multiple other proteins were detected, including non-E. coli ones as well (e.g. human keratins, chicken lysozyme used during cell lysis, proteases used in the MS analyses), while BcsB peptides were detected as contaminant in almost every excised band

(which is one of the reasons to perform epitope tagging and western blot analyses on multiple subunits in parallel).

The listed contaminants are proteins consistently identified in multiple experiments, through a significant number of peptides, and with molecular weights approximately corresponding to the running behavior on the gel. All proteins are clear membrane proteins and/or are known to co-purify during FLAG tag-dependent purification of inner-membrane proteins. The full protein names, the above clarification, as well as the appropriate references are now included in the affected figure legends (lines 446-451; 469-475). Their pull-down is likely due to non-specific interactions with the resin, as their amount tends to increase in conditions where specific interactions are compromised (especially in the delta-BcsQ background).

4. In the later experiments, a modified operon is used with the goal of overexpression. Was the complex really overexpressed in these conditions, relative to the WT format? If so, to what extent?

We now present data on the relative levels of expression from 1094 *bcsA^{HA-FLAG}*, 1094-2KT-*bcsA^{HA-FLAG}* and BL21(DE3) *bcsEFG+bcs^{His}RQA^{HA-FLAG}B* backgrounds. We also present data showing calcofluor fluorescence in the various genetic backgrounds (Extended Data Fig. 3b-c).

5. Are cellulose synthase activity values for the various subcomplexes?

Synthase activity is presented for 6 different complexes (Extended Data Fig. 7d). As this is an endpoint assay, we can only conclude that expressed BcsA protein is catalytically active in all contexts but it is the level of expression, assembly and stability of the system that are affected in the mutant *bcsQ* backgrounds.

6. Is it possible to generate any sense of the stoichiometry of the various proteins from the stained gel profiles and do the homotypic interaction results help?

Determining the exact stoichiometry has proven very challenging as bands for multiple components as well as for co-eluting contaminants tend to overlap. We have therefore refrained from presenting such data.

7. Can you please explain the origin of the eclectic detergent mixture used to extract the complex – is it from a screen or past precedent (in which case, please add a reference)?

For the initial detection of complex formation we used 0.1% Triton X-100, i.e. relatively mild and standard membrane solubilization conditions (line 744). The eclectic detergent mixture used for structural studies has been empirically optimized for this specific system. Nevertheless, we have relied as a starting point on studies on similar multicomponent secretion systems (T4SS, T6SS), the references for which have been added to the methodological section of the manuscript (lines 828-829).

Editorial Comments:

Line 55: I don't think "unique" is needed here; it might be an architecture not seen before but we don't yet know if will be unique.

We thank the reviewer and have substituted "unique" with "previously unobserved" (line 41).

Line 72: add a ref for the statement about BcsE.

We apologize for omitting the appropriate reference here and have included it in the revised version. The reference is Fang *et al.* and is cited elsewhere in the text as well (line 81; line 188).

Fig 1C: I think the calcofluor colonies need a negative control for non-experts. I know these controls appear in Extended Data Fig 1 but some readers may not want to dig that deep.

We fully agree with the reviewer and have now included a negative control to the figure, Δbcs , where the entire *bcs* cluster is deleted without polar effects (Figure 1c, Extended Data Fig. 1c).

Line 81: please define “crude” solubilization. I assume this is the same protocol described in extended methods?

We have removed “crude” from the sentence. The conditions for solubilization used for the initial detection of the Bcs interacting partners and for the complex purification for structural studies are detailed in the methods section.

Line 100: please define “mild” chemical crosslinking.

The crosslinking conditions are described in the methods section. We have added a sentence briefly describing the main principles of the GraFix technique to the main text as well (lines 110-113).

Line 108: I do not think “remarkably” is needed here. That this is a new structure shouldn’t be surprising and the significance of the finding is clear.

We agree and have removed the adverb from the revised text.

*Line 120: does BL21 lack the *bcs* genes, or is it unable to make cellulose because of a mutation?*

BL21 cells do not lack the *bcs* genes but are in fact unable to secrete cellulose, now shown in the extended data (Extended Data Fig. 3b). In fact, there are multiple SNPs in the essential for cellulose secretion genes *bcsQ*, *bcsB* and *bcsC* but their specific effects require further investigation. However, we were unable to rescue cellulose secretion even after overexpression of the complete set of *bcs* genes (Extended Data Fig. 3b), indicating that B-strain derivative strains might have evolved different regulatory pathways to bias extracellular matrix production toward use of alternative polymers (e.g. BL21 (DE3) cells readily secrete curli upon elevated c-di-GMP levels).

We have also modified the text to underscore the added data (lines 137-140; 262-266).

Line 145: another example where “remarkably” seems unnecessary.

Again, we agree with the reviewer and have removed the adverb.

Extended Data Fig. 7. Please explain in the legend the relevance of the Zip controls for non-experts.

This corresponds to the positive control described in the original Karimova *et al.* reference on the method. The colonies are co-transformed with the pT25-ZIP and pT18-ZIP plasmids, which encode adenylate cyclase fragments fused to the homodimerizing leucine zipper regions of yeast protein GCN4. We have modified the figure legend to clarify the control (lines 527-529).

Extended Data Fig. 8b. This can be updated with the T2SS translocon structure that just appeared in NSMB.

We have added both the latest studies on T2SS and T7SS to the figure, as well as the corresponding references (Extended Data Fig. 9b).

Reviewers' comments:

Reviewer #1 (Remarks to the Author):

It is somewhat surprising that the handedness could not be determined. In the interest of the authors they are advised to demonstrate the correctness of the structure (handedness).

- It is expected that the 2D reprojections are similar to the 2D projections used as inputs for the 3D reconstitutions. However, the authors should show a series of 2D projections that were left out during the 3D reconstitution process but could be found by projecting the 3D map into the direction of the respective Euler angles
- Did the authors try the tilt-validation server to test the handedness (RELION has a 50% chance to arrive at the correct handedness) - if so, what was the outcome?

Reviewer #2 (Remarks to the Author):

This is an elegant piece of work that adds considerable insight to the field.

The authors have addressed my previous concerns/comments by either modifying the manuscript or providing justifiable rationales for their arguments.

One minor point that needs to be address is that the CPS system in Extended Data Fig. 9 is the only one that is represented in cartoon/ribbon representation, but an $\sim 3\text{\AA}$ resolution structure of the T2S secretin as well as a near atomic resolution structure of the T3S basal body are now available and it would be disingenuous not to include these here.

The authors also need to carefully review the manuscript to ensure all minor typographical mistakes and grammar have been corrected is necessary.

For example:

- Line 245-6 "and their binary interactions with the essential for the system's assembly" does not make grammatical sense.
- BL21 sometimes has a lower case l

Reviewer #3 (Remarks to the Author):

The authors have appropriately addressed all of the concerns I raised about the original submission.

Reviewers' comments:

Reviewer #1 (Remarks to the Author):

“It is somewhat surprising that the handedness could not be determined. In the interest of the authors they are advised to demonstrate the correctness of the structure (handedness).”

We strongly disagree that the terms handedness and correctness can be used synonymously. As we have demonstrated experimentally in our original submission and in the revised manuscript, all independent initial models based on samples coming from different expression systems, data collected on different datasets and even using different microscopes, and analysed using different methods for initial model generation, refined independently to the same final 3D model with virtually the same features. This excludes bias that can stem from an incorrect input model and strongly supports overall correctness of the structure.

Moreover, our exhaustive and independent structural analyses on multiple sub-complexes demonstrated that we consistently identify the corresponding densities within that of the Bcs macrocomplex, pointing again towards correctness of the integrated 3D model.

We made clear in the manuscript that uncertainty remains concerning the handedness of the structure. However, the handedness of the structure does not change any of the interpretation of the structural assembly and its relation to oligomeric assembly, mechanism and stability of the cellulose secretion system, to which we have, to the various reviewers' agreement, made an unprecedented and significant contribution. In particular, models of each hand are mere mirror images of each other so the structural features, together with the possible fitting of the cellulose synthase BcsAB subunits, orientation relative to the bacterial envelope, oligomeric assembly, stability effects or any of our other subsequent and exhaustive analyses and their functional implications in bacterial physiology, secretion and biofilm formation would be virtually unchanged.

Addressing referee 1's comments would lead to more unreasonable delays, annihilating the impact of our commitment without improving our significant contribution to the field.

“ - Did the authors try the tilt-validation server to test the handedness (RELION has a 50% chance to arrive at the correct handedness) - if so, what was the outcome? “

As we underscored previously, we have opted to low-pass filter our data in order to decrease noise artefacts that can affect our input models. This limits us to very low resolution of the initial models (60Å) at which the asymmetry and handedness are not evident. In fact, the features that determine those are refined in the later rounds of reiterative refinement (50 rounds in total), which integrate all tens of thousands of particles and the full range of low and higher resolution data, and where the signal-to-noise ratio is dramatically improved.

We agree with the reviewer that Relion has 50% chance to find the correct handedness and since in some final refined models the handedness is flipped, we have opted to be completely open and to allow the caveat that the actual handedness can be reversed relative to the more consistent final model and that that would be settled in downstream studies by us or others.

We have not resorted to the specific tilt validation server mentioned by the reviewer as it only inputs single particles, requires the use of specific software and by the words of its own developers is still being developed and tested (<https://www.ebi.ac.uk/pdbe/emdb/validation/tiltpair/>). Moreover, a brief review of recent publications in Nature Communications reveals multiple studies where there is no use of this server or method (e.g. Frigola et al. 2017, doi:10.1038/ncomms15720 ; Ukleja et al. 2017, doi:10.1038/ncomms10433) so it cannot be required.

Rather than using the server, we employed random conical tilt reconstruction, which relies on higher SNR class averages and not isolated particles, and obtained the same final structural model, again with properly acknowledged ambiguity in the handedness (we should underscore that RCT was also requested by this reviewer in round 1).

In all fairness, we did implement the tilt-pair validation method in EMAN2.0 but program-matching of single particles to 2D reprojections was inconsistent likely due to stain artefacts, low SNR, and defocus gradient in tilt leading to inapplicable CTF correction. We would therefore not report those data as reliable.

Most importantly, and as mentioned above, the handedness does not change any of the interpretation of the structural assembly and its relation to assembly, mechanism and stability of the cellulose secretion system, to which we have, to the various reviewers' agreement, made an unprecedented and significant contribution. In particular, models of each hand are mere mirror images of each other so the structural features, together with the possible fitting of the cellulose synthase BcsAB subunits, orientation relative to the bacterial envelope, oligomeric assembly, stability effects or any of our other subsequent and exhaustive analyses and their functional implications in bacterial physiology, secretion and biofilm formation would be virtually unchanged.

Bottom line, the level of uncertainty here, is equivalent to looking at oneself in the mirror and doubting who that person might be.

- It is expected that the 2D reprojections are similar to the 2D projections used as inputs for the 3D reconstitutions. However, the authors should show a series of 2D projections that were left out during the 3D reconstitution process but could be found by projecting the 3D map into the direction of the respective Euler angles

Regarding reprojections in general, we agree that 2D projections used for 3D reconstruction are expected to be retrieved by reprojecting back the final model and actually that is by itself validation of the refinement procedure.

However, as in extended data fig. 4 we are comparing class averages and not single 2D projections to back-projections of the final refined 3D model, we would like to emphasize that - contrary to some earlier-generations software packages - Relion performs 3D classification and refinement on the ensemble of single particles rather than using and restraining those particles to specific 2D class averages to be input for building the 3D model.

Therefore, the comparison shown in the figure is not showing "2D reprojections used as inputs for the 3D reconstitutions", but rather that reprojections of the refined model match a large number of single particles. So we find that the reviewer's comment is somewhat technically misleading. We would also like to emphasize that we only included this figure based on the reviewer's own request at round 1, so being criticized for it by the same reviewer is certainly confusing and unacceptable.

However and most importantly to this concern, we obtain the same class averages again and again, and again using different datasets and even samples (1094- vs. B121(DE3)- purified complex with (i) labeled BcsE, (ii) unlabelled BcsE, and (iii) no BcsFG (figure 2, figure 3 and supplementary fig. 6), so indeed we have already demonstrated explicitly that reprojections match even particles omitted in the initial analyses used for a specific 3D reconstruction and retrieved in completely independent experiments, from differently sourced sample, data collected on different microscopes, for different versions of the submission, etc. So again, we strongly believe that there are no outstanding concerns to be further addressed.

Reviewer #2 (Remarks to the Author):

This is an elegant piece of work that adds considerable insight to the field.

The authors have addressed my previous concerns/comments by either modifying the manuscript or providing justifiable rationales for their arguments.

We thank the reviewer 2 for these comments.

One minor point that needs to be address is that the CPS system in Extended Data Fig. 9 is the only one that is represented in cartoon/ribbon representation, but an $\sim 3\text{\AA}$ resolution structure of the T2S secretin as well as a near atomic resolution structure of the T3S basal body are now available and it would be disingenuous not to include these here.

We carefully considered this suggestion to include ribbon models in T2SS and in the T3SS base in extended Data figure 9. However, the point of this figure is to provide a global idea of the organization of known bacterial secretion systems. While adding ribbon diagrams will not provide much additional information to this respect, including crystal structures for components of all other secretion systems will make the figure overcrowded and poorly legible. Instead, we added a sentence to the legend, mentioning the availability of higher resolution models for sub-complexes and individual subunits for most of the systems.

The authors also need to carefully review the manuscript to ensure all minor typographical mistakes and grammar have been corrected is necessary.

For example:

- Line 245-6 "and their binary interactions with the essential for the system's assembly" does not make grammatical sense.*
- BL21 sometimes has a lower case l*

We went through the manuscript and performed the requested corrections in the text and in the figures

Reviewer #3 (Remarks to the Author):

The authors have appropriately addressed all of the concerns I raised about the original submission.

We thank the reviewer 3 for this comment.